# Toward Efficient Low-Precision Training: Data Format Optimization and Hysteresis Quantization

**Sunwoo Lee, Jeongwoo Park, Dongsuk Jeon**
Graduate School of Convergence Science and Technology
Seoul National University, Seoul, Korea
{ori915,jeffjw,djeon1}@snu.ac.kr

## Abstract

As the complexity and size of deep neural networks continue to increase, low-precision training has been extensively studied in the last few years to reduce hardware overhead. Training performance is largely affected by the numeric formats representing different values in low-precision training, but finding an optimal format typically requires numerous training runs, which is a very time-consuming process. In this paper, we propose a method to efficiently find an optimal format for activations and errors without actual training. We employ this method to determine an 8-bit format suitable for training various models. In addition, we propose hysteresis quantization to suppress undesired fluctuation in quantized weights during training. This scheme enables deeply quantized training using 4-bit weights, exhibiting only 0.2% degradation for ResNet-18 trained on ImageNet.

## 1 Introduction

Deep neural networks have been used in various fields such as vision, audio, natural language processing, and reinforcement learning. As larger and more complex neural networks are adopted, the energy and time consumed for training have become a critical issue in hardware implementation. Using low-bit representations in training significantly reduces hardware overhead and memory footprint; hence, neural network training with limited precision has been extensively studied recently. For instance, 16-bit formats are already adopted in commercial devices such as FP16 (IEEE, 2019) in Nvidia GPUs and bfloat16 (Kalamkar et al., 2019) in Google TPU (Wang et al., 2019). Also, Köster et al. (2017) suggested a new data format using a shared exponent suitable for low-precision training. Recently, it has been demonstrated that even 8-bit formats could be adopted in deep neural network training with reasonable accuracy (Sun et al., 2019; Fox et al., 2020). However, there are various issues in realizing low-precision training in practical applications as detailed below.

**Optimal data format for low-precision training**: Training performance is susceptible to the data format we use to represent variables in the network. When a value is represented using a floating-point format with a fixed number of bits, there is a trade-off between dynamic range and precision. For instance, allocating more bits to the exponent part in a floating-point format enlarges the dynamic range but lowers precision due to fewer bits in the mantissa part. Recent studies on 8-bit training suggest various ways to reduce the dynamic range required for number representation to enhance representation precision. Early work on 8-bit training (Wang et al., 2018) adopts a 5-bit exponent to represent different variables using a single format, but Sun et al. (2019) examine the statistics of each variable and optimize the numeric formats separately. Specifically, the values used in the forward path (weight and activation) have a relatively narrow dynamic range, and only 4 bits are allocated to the exponent. Fox et al. (2020) propose to divide data into smaller blocks and assign a shared exponent bias to each block. Since the values in a block tend to exhibit similar statistics, the forward (weight and activation) and backward (error) paths could be represented using only 2-bit and 4-bit exponents, respectively. Note that the shared exponent bias is effectively identical to the scaling factor. If a variable has a value of $m \cdot 2^e$ and a shared exponent bias of $b$, then its actual value is $m \cdot 2^{e+bias}$, which is identical to the scaling factor of $2^{bias}$. However, these approaches are difficult to generalize since we should empirically decide numeric formats for each task, neural

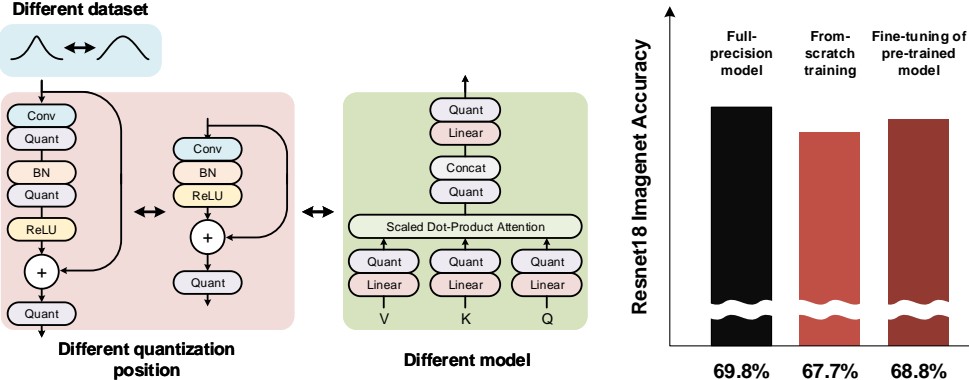

(a) Various quantized training environments

(b) Performance degradation in quantized models

Figure 1: Two Challenges in low-precision training. (a) An optimal format varies with dataset, quantization scheme, and model. (b) There is a performance gap between from-scratch training with quantized weights and fine-tuning of a pre-trained full-precision model.

network structure, and quantization scheme (Fig. 1). Furthermore, analyzing the statistics of each variable is not enough to determine an optimal format. Their distributions often have a long tail, and hence the dynamic range of the numeric format should be experimentally selected through many trial-and-errors in actual training.

**Performance degradation in from-scratch training**: Previous studies on quantized models show that a model could achieve comparable accuracy to full-precision models even using 1- or 2-bit weights (Choi et al., 2019; Martinez et al., 2020) through fine-tuning a pre-trained model. However, in low-precision training where a neural network is trained from scratch using low-precision values and computations, the trained model typically shows a noticeable accuracy drop (Elhoushi et al., 2021). Fig. 1(b) shows the Top-1 validation accuracy of ResNet-18 (He et al., 2016) trained on ImageNet (Deng et al., 2009) for different training schemes. The weights are quantized into a 4-bit base-2 logarithmic format. From-scratch training of the model with quantized weights results in a 2.1% accuracy drop, whereas only 1.0% degradation is observed if we fine-tune a pre-trained model. This suggests that even though a better solution (i.e., a set of parameters) exists for a given format, it cannot be reached through from-scratch training.

To formalize the issues above, here we divide quantization in low-precision training into two types: network quantization and data flow quantization. Network quantization refers to the quantization of the neural network model. An example of this type of quantization is weight quantization. In network quantization, we need to reduce the performance difference between from-scratch training and fine-tuning (Yang et al., 2019b). On the other hand, data flow quantization refers to the on-the-fly quantization that occurs when data propagate through the network in low-precision training. Examples include activation, error, and weight gradient quantizations. Additional errors are introduced in weight update computation due to this type of quantization, which leads to performance degradation. Hence, we need to find an optimal format to minimize accuracy drop due to computation errors in data flow quantization.

In this paper, we present a systematic approach to implementing low-precision training on various models and tasks. First, we present a method to efficiently find an optimal format for data flow quantization. In addition, we introduce a hysteresis quantization technique, a new quantization method for network quantization that can mitigate the issues of from-scratch training. Our main contributions are:

- We present **a method that can predict the training performance of various numeric formats for data flow quantization**. This method allows us to determine an appropriate data format for different neural network structures, datasets, and tasks efficiently.

- Using the method above, we propose **an optimal 8-bit format** suitable for low-precision training of various models, which enables quantization of BatchNorm layer input and improves hardware efficiency with minimal performance degradation.

- We propose **a new quantization scheme that utilizes the hysteresis effect** to improve the performance of from-scratch training in network quantization. This scheme enables ultra-low-precision training using 4-bit logarithmic weights.

## 2 DATA FLOW QUANTIZATION

### 2.1 NUMERIC FORMATS

There are many numeric formats that can be constructed with *n* bits depending on how much dynamic range is required and how many valid bits are used for representing a value. For example, using 8 bits we could implement 8-bit fixed point integer format, 8-bit floating-point formats such as FP152, FP143, and FP125 (FP1*xy* represents 1 sign bit, *x* exponent bits, and *y* mantissa bits), 8-bit posit format (Gustafson & Yonemoto, 2017), and 8-bit float-fix format (Han et al., 2019). Since the diversity of formats that could be formulated using *n* bits is nearly unlimited, here we assume some constraints to limit candidates while still including widely used formats such as fixed-point and floating-point formats as below:

- The MSB (Most Significant Bit) is used as a sign bit and other bits represent magnitude. Accordingly, only symmetric formats that have identical representable ranges for positive and negative numbers are considered. Two's complement representation is slightly asymmetric since it can represent one more negative value, but it does not incur a significant difference.

- The number of valid bits of a larger value is greater than or equal to the number of valid bits of a smaller value. The valid bits stand for significant digits in binary representation.

- The ratio between consecutive representable values does not exceed 2. For example, the base-4 logarithmic format is excluded.

We could obtain 166 8-bit formats that meet these constraints. Then, we reduce 1 and 2 valid bits in each format to obtain 7- and 6-bit formats, resulting in 498 formats in total. More information on the numeric formats considered in our experiments is provided in Appendix A.1.

### 2.2 ACTIVATION AND ERROR QUANTIZATION

In a neural network consisting of *n* layers, the training process is described by

$$A_{l+1} = f_l(W_l^t, A_l) \tag{1}$$

$$E_l = g_l(W_l^t, E_{l+1}) \tag{2}$$

$$G_{wl} = h_l(A_l, E_{l+1}) \tag{3}$$

$$W_l^{t+1} = o(G_{wl}, W_l^t) \tag{4}$$

where $A$, $E$, $W$, and $G_w$ are activation, error, weight, and weight gradient, respectively. $f$, $g$, $h$, and $o$ are forward, backward, gradient, and update functions. $l$ and $t$ represent the layer number and time step. We follow the quantized training scheme suggested by Fox et al. (2020), but with the following modifications to reduce hardware implementation costs. $A$ and $E$ are quantized not only for the GEMM input but also for the BatchNorm layer input. BatchNorm layer normalizes input using the mean and variance of each channel, but these values are obtained only after observing all the inputs from the previous layer, necessitating that all input values are temporarily stored in memory. Therefore, quantizing the BatchNorm layer's input significantly reduces memory footprint and memory access overhead. Additionally, the scope of sharing exponent bias is extended to a layer ($A_l$ and $E_l$) to avoid the overhead of aligning partial sums from different blocks in block-wise exponent sharing. Finally, instead of determining the shared exponent bias by analyzing all values in the layer, we conservatively update it by detecting overflow and underutilization that occurred in the previous mini-batch.

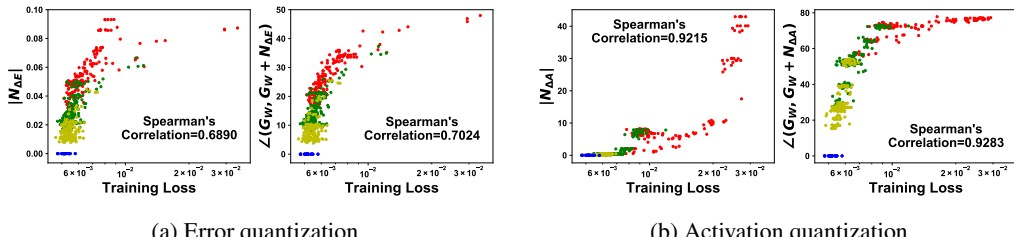

(a) Error quantization             (b) Activation quantization

Figure 2: Training loss vs. proposed performance indicators. Blue dots represent full-precision training runs, and yellow, green, and red dots represent training runs with 8-, 7-, and 6-bit formats, respectively.

## 2.3 INDICATORS OF TRAINING PERFORMANCE

**Effect of quantized error**: Quantizing the error $E$ in the backward path is independent of how the forward path behaves since the loss surface of the model does not change. Therefore, the optimal $W$ that the network needs to reach through training remains the same regardless of the error quantization scheme. However, when the error is quantized, a quantization error $\Delta E$ is introduced in $E$, which incurs a noise $N_{\Delta E}$ in $G_w$ through the gradient function in Eq. 3 and potentially updates each weight in the wrong direction. While some amount of noise may improve the training performance through regularization, using low-precision formats already introduces a large noise in the network, incurring performance degradation (see Appendix A.8). Therefore, we suggest that the weight gradient error $N_{\Delta E}$ could be a good indicator of degradation in training performance. One way to implement this is predicting performance using the magnitude of $N_{\Delta E}$; however, if the noise is in the same direction as $G_w$, it would only change the amount of each update and result in a less severe effect. Instead, we could measure the misalignment between $G_w + N_{\Delta E}$ and $G_w$ for performance prediction. The misalignment between two vectors is estimated by

$$\angle(A, B) = \cos^{-1}\left\{ \frac{A \cdot B}{\|A\|_2 \cdot \|B\|_2} \right\} \tag{5}$$

Then, the change in the update direction due to $N_{\Delta E}$ is $\angle(G_w, G_w + N_{\Delta E})$. We can expect that the smaller $\angle(G_w, G_w + N_{\Delta E})$, the better the training performance.

**Effect of quantized activation**: Contrary to error quantization, activation quantization affects the way the forward path operates, and the loss surface of the model changes. Hence, the global optima of weight parameters shift, where the amount of shift would be proportional to the quantization noise. The displacement of global optima can be indirectly estimated using the direction of the weight gradients $G_w$. If the angle $\angle(G_w, G_w + N_{\Delta A})$ is small, the deviation of the global optima is expected to be small as well, suggesting a better training performance.

In the discussions above, we assumed that the angles $\angle(G_w, G_w + N_{\Delta E})$ and $\angle(G_w, G_w + N_{\Delta A})$ could be used to predict training performance. We experimentally prove this by comparing the training performance of different numeric formats. For 498 numeric formats in 6 to 8 bits, we compare the loss obtained from training with the proposed performance indicators (Fig. 2). Training loss is obtained by training ResNet-18 on CIFAR-10 dataset using SGD with a momentum of 0.9 for 60 epochs. The batch size is 128 images and the initial learning rate is 0.1, which is decayed by a cosine scheduler. We average angles from 100 mini-batches after quantizing a pre-trained model. Note that we use $G_w$ of the first layer since it can reflect quantization errors that occur in the activations and errors of all the layers in the network. The weight gradients from the full-precision network, the network with quantized activations, and the network with quantized errors are $G_w$, $G_w + N_{\Delta A}$, and $G_W + N_{\Delta E}$, respectively. Fig. 2 shows that using the misalignment angle results in not only a higher Spearman's correlation but also a more distinct shape for low training losses, making it a better metric than the error magnitude. For instance, using the error magnitude would predict the best format for transformer incorrectly (see Fig. 8(e) in Appendix A.3). While obtaining the misalignment angle requires additional computations, its overhead is negligible since the part that requires the most time and computation is to obtain $G_w$, $G_w + N_{\Delta E}$, and $G_w + N_{\Delta A}$, which is still significantly lower than actual training. Using this method, we could determine the optimal format for a specific neural network model, dataset, and task very efficiently as we only need to

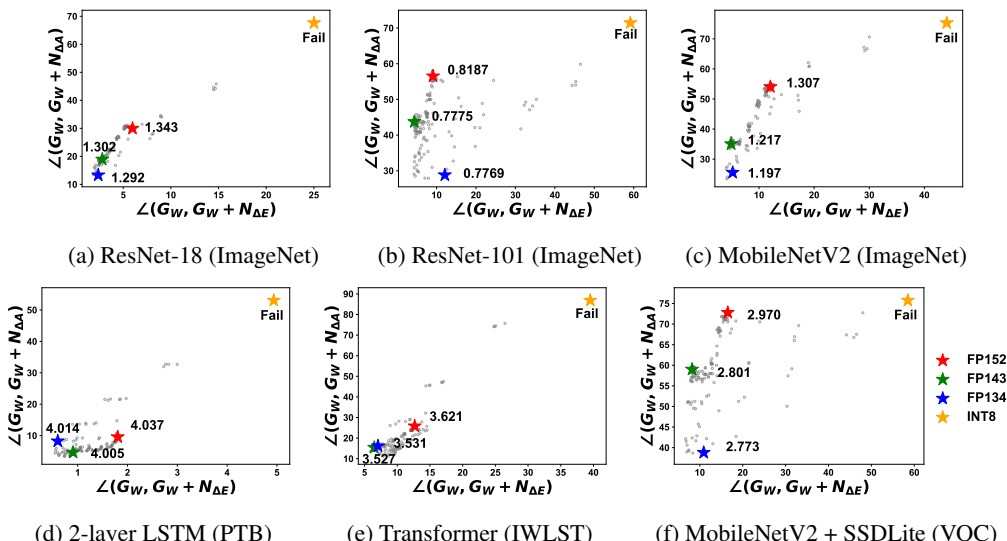

Figure 3: Scatter plots displaying misalignment angles for 166 8-bit formats. The numbers next to four selected data formats (INT8, FP152, FP143, and FP134) represent the loss measured in actual training.

measure the misalignment angle without time-consuming network training. For experiments in Fig. 2, the amount of computation is reduced by 99.6%, and the reduction will be even larger for larger datasets and complex networks that need more epochs for training.

## 2.4 OPTIMAL FORMAT FOR DATA FLOW QUANTIZATION

Here we show that we could find an optimal format for training with quantized errors and activations using the proposed performance estimation method above. To find a format suitable for a wide range of models, we select six models with different architectures, layer types, and target tasks that are widely used in quantized training research for experiments: ResNet-18, ResNet-101, MobileNetV2 (Sandler et al., 2018), 2-layer LSTM, small transformer for translation on the IWSLT German to English dataset (Cettolo et al., 2014), and SSD-Lite (Liu et al., 2016) with MobileNetV2. We first measure misalignment angles for 166 8-bit formats. To verify the correlation between the training performance and the misalignment angles, we select four formats that exhibit low hardware implementation costs (INT8, FP152, FP143, and FP134) and train the networks using each format. While we may use different formats for activation and error, it requires a complicated datapath (Sun et al., 2019) and hence we only consider a single format for both variables. The experimental results in Fig. 3 demonstrate that the training performance is higher if both misalignment angles are small in all tasks and models, confirming that the proposed indicators could be used to determine the optimal numeric format. Fig. 3 suggests that FP134 and FP143 are the best candidates across all models. For hardware implementation, FP134 is the most optimal format due to its low implementation cost, which is discussed in Appendix A.7 in detail. Note that using the error magnitude leads to the same conclusion that FP134 is the best format for the target models. See Appendix A.3 for more details.

## 3 NETWORK QUANTIZATION

In quantized neural networks, the weight parameters are generally quantized in a way that minimizes the quantization error (Choi et al., 2019; Martinez et al., 2020). For instance, if $x$ is quantized into a fixed-point format through $s \times \text{round}(\frac{x}{s})$, a proper value is selected for the scaling factor $s$ to minimize the quantization error. However, as the weights continue to change during training, we need to calculate $s$ for every update, which could cause significant overhead. Therefore, prior studies on low-precision training suggest constraining the scaling factor to the power of 2 in the shared exponent (Köster et al., 2017) or the shared exponent bias (Fox et al., 2020). In this section,

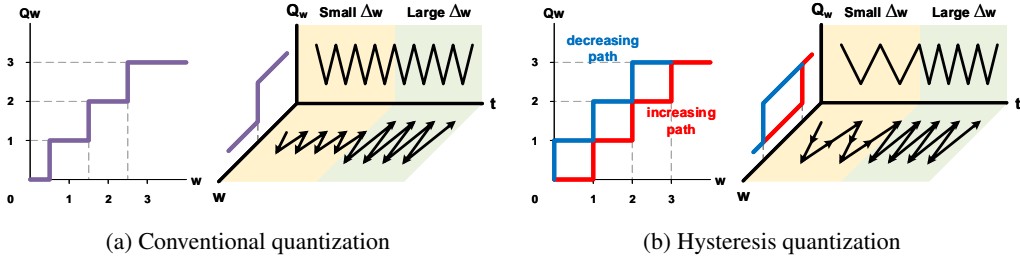

(a) Conventional quantization          (b) Hysteresis quantization

Figure 4: Comparison of quantization schemes.

we analyze the issues behind weight quantization and propose a new quantization scheme to mitigate those issues.

### 3.1 FLUCTUATION OF WEIGHT PARAMETERS

In typical low-precision training, a master copy of weight parameters is separately maintained in high precision, and those weights are updated based on the computed weight gradient. This high-precision weight is quantized into a low-precision format and used for the forward path computation during training. If the scaling factor $s$ is constrained to $2^n$, the quantization threshold remains the same unless $s$ is updated due to overflow or underutilization. If the optimal weight is located between two representable values of a data format, the quantized weight would fluctuate alternately between the two values in each update (Fig. 4(a)) even for a very small weight update, causing large fluctuations and undermining training performance.

### 3.2 HYSTERESIS QUANTIZATION

To mitigate the fluctuation issue above, we propose to introduce the concept of hysteresis to quantization. More specifically, we quantize each weight differently in a way that the quantized value tends to stay at its current value, effectively minimizing undesired oscillation between two values due to small weight updates. The equation below shows an example of the proposed quantization scheme.

$$Q_w^t = \begin{cases} \lfloor w^t \rfloor, & \text{if } w^t > Q_w^{t-1} \\ \lceil w^t \rceil, & \text{if } w^t < Q_w^{t-1} \end{cases} \tag{6}$$

where $w$ is the original value, $Q_w$ is its quantized value, and $t$ is the time step. The proposed hysteresis quantization reduces fluctuation significantly, stabilizing the training process and allowing the network to reach global optima more efficiently. In Fig. 4(b), if the weight change $\Delta W$ is small, then enough number of those changes should be accumulated to flip $Q_w$. Hence, the update frequency is now proportional to the weight gradient. This helps the network to learn better while suppressing fluctuations for small $G_w$ values. Alternatively, we may mitigate weight quantization errors by adopting AdaRound (Nagel et al. (2020)), which learns whether each weight should be rounded up or down to produce the same output as high-precision weights. However, whenever full-precision weights are updated, we need to re-train the learnable parameters (i.e., quantization scheme of each weight), incurring a large overhead and undermining the benefit of low-precision training.

### 3.3 ULTRA-LOW-PRECISION FORMAT FOR NETWORK QUANTIZATION

To verify the effectiveness of the proposed hysteresis quantization, we select 4-bit logarithmic representation as an ultra-low-precision format for weight parameters. This format has the same dynamic range as INT8 which is widely used for weight quantization, and is more hardware-efficient as multiplication is implemented only using simple shift operations. There have been attempts to use logarithmic weights in quantized neural networks (Lee et al., 2017; Elhoushi et al., 2021), but from-scratch training shows a significant performance degradation. In logarithmic data formats, the interval of quantization points is not uniform, making the effect of fluctuation more severe.

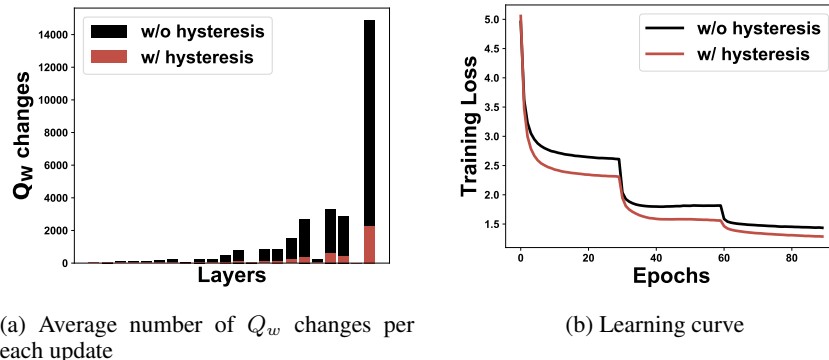

(a) Average number of $Q_w$ changes per each update

(b) Learning curve

Figure 5: Experimental results of hysteresis quantization.

Fig. 5 shows experimental results of ResNet-18 training on ImageNet using 4-bit logarithmic weights. Note that we apply channel-wise quantization to the convolutional layers to compensate for the insufficient expression range and layer-wise quantization to the other types of layers. Further details on the experimental setup are provided in Appendix A.5.1. First, we measure how many quantized weights $Q_w$ change when the network performs one weight update using a mini-batch and average them over the first 100 updates in the 60th epoch. The experimental result displayed in Fig. 5(a) clearly shows that using hysteresis significantly reduces weight change frequency and stabilizes the training process. Fig. 5(b) compares the training performance of quantization schemes with and without hysteresis. Hysteresis quantization not only speeds up training but also achieves better results at the end of training. Note that hysteresis quantization is applicable to other data formats, and additional experimental results can be found in Appendix A.4.

## 4 EXPERIMENTAL RESULTS

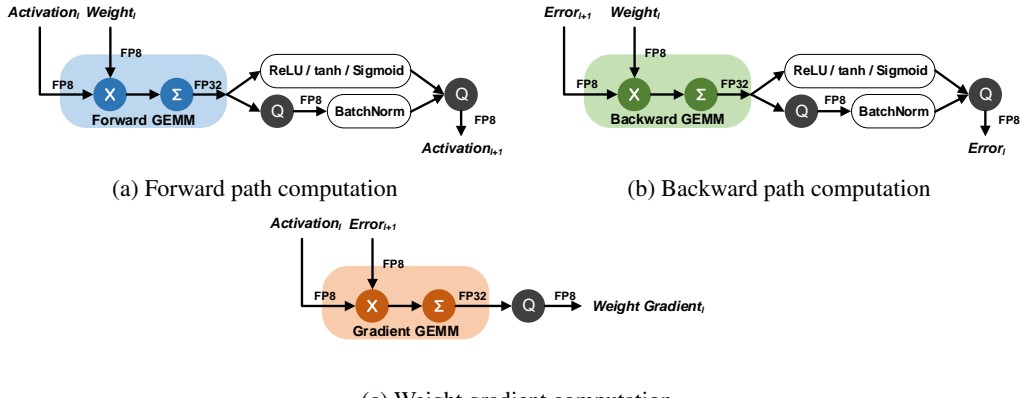

(a) Forward path computation

(b) Backward path computation

(c) Weight gradient computation

Figure 6: Computation flow of 8-bit low-precision training.

### 4.1 LOW-PRECISION TRAINING SCHEME

For low-precision training, we need to quantize four variables: activation, error, weight, and weight gradient. In our experiments, we apply the quantized training scheme detailed in 2.2 to all of these variables, as depicted in Fig. 6. As in previous studies on 8-bit training, the inputs of GEMM are all quantized into 8 bits. Additional functions are applied to GEMM results in the forward and backward paths. ReLU, tanh, and sigmoid functions are performed directly on the input, whereas the input of BatchNorm is re-quantized.

Table 1: Training Performance of FP134 Data Format

| Model (Dataset) [Metric] | Baseline (FP32) | FP134 |
|---|---|---|
| ResNet-18 (ImageNet) | 69.8 | 69.8 |
| ResNet-101 (ImageNet) | 77.6 | 77.4 |
| MobileNetV2 (ImageNet) | 72.2 | 71.9 |
| 2-layer LSTM (PTB) [ppl.] | 91.5 | 92.0 |
| Transformer (IWSLT) [BLEU] | 34.8 | 34.5 |
| MobileNetV2 + SSDLite (VOC) [mAP] | 68.3 | 68.2 |

Table 2: Comparisons of Data Formats for Low-Precision Training of ResNet-18 on ImageNet

| Quantization Scheme | Formats (Exponent, Mantissa) | | | | | | Top-1 Accuracy | |
|---|---|---|---|---|---|---|---|---|
| | w | GEMM Input x | Batch -Norm Input x | dw | dx | Acc. | FP32 | Proposed |
| SWALP (Yang et al., 2019a) | $8^1$ | $8^1$ | - | $8^1$ | $8^1$ | $32^1$ | 70.3 | 65.8 |
| S2FP8 (Cambier et al., 2020) | (5,2)/(8,23) | (5,2) | - | (5,2) | (5,2) | (8,23) | 70.3 | 69.6 |
| HFP8 (Sun et al., 2019) | (4,3) | (4,3) | (6,9) | (6,9) | (5,2) | (6,9) | 69.4 | 69.4 |
| BM8 (Fox et al., 2020) | (2,5) | (2,5) | $31^1$ | (6,9) | (4,3) | $31^1$ | 69.7 | 69.8 |
| FP8-SEB (Park et al., 2021) | (4,3) | (4,3) | (4,3) | (4,3) | (4,3) | (6,23) | 69.7 | 69.0 |
| FP134 (Ours) | (3,4) | (3,4) | (3,4) | (3,4) | (3,4) | (6,23) | 69.8 | 69.8 |

[1] Fixed Point

## 4.2 8-BIT LOW-PRECISION TRAINING

In Section 2.4, we found that FP134 is the optimal format for low-precision training using the proposed performance prediction method. We measure the training performance of this format and compare it against other 8-bit data formats from recent studies by applying those formats to the training of various neural network models. More details on the experimental setup are provided in Appendix A.5. The performance of the proposed data format is summarized in Table 1. Overall, 8-bit training using FP134 achieves nearly the same performance as the full-precision training on all models. Even in MobileNetV2, which is known to be sensitive to quantization due to the small number of parameters, only 0.3% degradation occurred. Sun et al. (2019) show that HFP8 also exhibits only 0.2% accuracy degradation in MobileNetV2 (71.81% vs. 71.61%), but they quantize Batch-Norm input into 16 bits instead of 8 bits, roughly doubling the memory access and computational complexity. Additionally, since the forward and backward paths employ different data formats, HFP8 is actually implemented using 9-bit MAC units in hardware (Agrawal et al., 2021). Table 2 compares the training performance of various data formats for ResNet-18 training. The columns w, x, dw, dx, and acc refer to weight, activation, weight gradient, error, and GEMM accumulation, respectively. Our FP134 format exhibits no accuracy drop compared to full-precision training. HFP8 (Sun et al., 2019) and BM8 (Fox et al., 2020) demonstrate similar performance, but they both use higher precision to represent BatchNorm inputs, and different formats are adopted in the forward and backward paths, necessitating complex computation units when implemented in hardware, as decribed above. In addition, BM8 assumes block-wise sharing of exponent bias, incurring additional overhead in memory access and data alignment. FP8-SEB (Park et al., 2021) addresses this issue by employing layer-wise exponent bias sharing and multi-way MAC units, but it results in a 0.7% accuracy drop for ResNet-18 training. Contrarily, our data format shows no performance degradation, while deeply quantizing BatchNorm inputs into the same format and allowing for a simple datapath by using an identical data format in the forward and backward paths.

## 4.3 ULTRA-LOW-PRECISION TRAINING WITH 4-BIT LOGARITHMIC WEIGHTS

Elhoushi et al. (2021) recently demonstrated that 4-bit logarithmic weights could be used for network quantization. Fine-tuning of a pre-trained model only showed 0.2% accuracy degradation, but

Table 3: Comparisons of Training Schemes Using 4-bit Logarithmic Weights for ResNet-18

| | From[1] Scratch | Formats (Exponent, Mantissa) | | | Top-1 Accuracy | |
| --- | --- | --- | --- | --- | --- | --- |
| | | w | x, dw, dx | Acc. | FP32 | Proposed |
| DeepShift-Q (Elhoushi et al., 2021) | X | (3,0) | - | - | 69.8 | 69.6 |
| | O | (3,0) | - | - | 69.8 | 65.3 |
| FP134 + 4-bit Log W | O | (3,0) | (3,4) | (6,23) | 69.8 | 67.7 |
| FP134 + 4-bit Log W + Hysteresis | O | (3,0) | (3,4) | (6,23) | 69.8 | 69.6 |

[1] X: fine-tuning of pre-trained models, O: from-scratch training

Table 4: 4-bit Logarithmic Weight Training with and without Hysteresis Quantization.

| Model (Dataset) [Metric] | FP32 | 4-bit Log W | 4-bit Log W + Hysteresis | Δ |
| --- | --- | --- | --- | --- |
| ResNet-18 (ImageNet) | 69.8 | 67.7 | 69.6 | +1.9 |
| ResNet-101 (ImageNet) | 77.6 | 76.6 | 77.3 | +0.7 |
| MobileNetV2 (ImageNet) | 72.2 | 67.5 | 69.6 | +2.1 |
| 2-layer LSTM (PTB) [ppl.] | 91.5 | 93.3 | 93.2 | -0.1[1] |
| Transformer (IWSLT) [BLEU] | 34.8 | 33.5 | 33.8 | +0.3 |
| MobileNetV2 + SSDLite (VOC) [mAP] | 68.3 | 63.3 | 66.1 | +2.8 |

[1] Lower is better

from-scratch training of the same model resulted in a 4.5% accuracy drop in ResNet-18 training (Table 3). Similarly, our experiments show 2.1% lower accuracy when training ResNet-18 using 4-bit logarithmic weights and FP134 format for other variables. However, using hysteresis quantization greatly improves the training performance and reduces accuracy degradation to 0.2%. This is identical to the training performance achieved through fine-tuning a pre-trained model by Elhoushi et al. (2021), confirming that hysteresis quantization effectively solves the issue of sub-optimal solutions in from-scratch training. In addition, Table 4 demonstrates that hysteresis quantization improves the training performance in all target models. Note that we quantized all trainable weights except for the BatchNorm parameters into 4 bits in experiments; the training performance could be further improved by using higher precision for error-sensitive parts such as the first/last layers and residual connections.

## 5 CONCLUSION

In low-precision training, the dynamic range of a tensor is data-dependant, and hence an optimal data format depends on various factors such as model, dataset, and quantization scheme. We showed that the training performance of a specific data format for activation and error could be predicted by observing the errors introduced in the weight gradients. Based on this observation, we determined an optimal 8-bit format for low-precision training very efficiently without running numerous training runs. The proposed FP134 format achieved a similar or better accuracy compared to prior works, while allowing for efficient hardware implementation through quantizing BatchNorm inputs and using a unified data format in both forward and backward paths. In addition, we proposed the hysteresis quantization scheme for network quantization, which improves training performance by suppressing undesired fluctuations and stabilizing the training process. In ultra-low-precision training with 4-bit logarithmic weights, hysteresis quantization significantly improves training performance by mitigating sub-optimal solutions, closely matching the performance obtained through fine-tuning a pre-trained model. We expect that these two schemes can complement each other to enable practical low-precision training on various models and tasks.

ACKNOWLEDGMENTS

This work was supported by the National Research Foundation of Korea (Grant No. NRF-2022R1C1C1006880). The EDA tool was supported by the IC Design Education Center.

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

# A   APPENDIX

## A.1   VARIOUS FORMATS ANALYZED IN SECTION 2

In this paper, we made three assumptions on the quantization formats that were analyzed. Firstly, 1-bit is allocated as a sign bit, so only symmetric formats are allowed, and secondly, the number of valid bits with a large absolute numerical value must be greater than or equal to the number of valid bits with a small absolute numerical value. Lastly, the base does not exceed 2.

Considering the above assumptions, we provide a systematical approach for generating different quantization methods that were used for analysis in Section 2, in order to create quantization methods that have trade-offs in terms of dynamic range and the number of valid bits. The quantization method is expressed with the following items: i) a list of decreasing positive real numbers $P$ that contains the interval points (Eq. 7) and ii) a non-increasing integer list $L$ that accompanies the interval list, with each item representing the number of valid bits (Eq. 8). Here, $s$ is shared exponent bias.

$$P = \{2^{s+1}, 2^s, 2^{s-1}, ..., 2^{s-K+1}\} \text{ where } s \in N \tag{7}$$

$$L = \{l_0, l_1, ..., l_{K-1}\} \text{ where } l_k \in N, i < j \Rightarrow l_i \geq l_j \tag{8}$$

The quantization points $Q$ are generated in each of the intervals that are sliced with $2^{l_k-1}$ evenly distributed datapoints. If the interval is $\{2^{s+1}, 2^s\}$, the quantization point $Q$ can be expressed by Eq. 9.

$$Q = \{2^s, 2^s(1 + \frac{1}{2^{l_k-1}}), 2^s(1 + \frac{2}{2^{l_k-1}}), ..., 2^s(1 + \frac{2^{l_k-1}-1}{2^{l_k-1}})\} \tag{9}$$

Notice that $L$ for an $\alpha$-bit quantization must satisfy

$$2^{\alpha-1} = 1 + \sum_{k=0}^{K-1} 2^{l_k-1} \tag{10}$$

Since the format is symmetric, only half of the data points are assigned to positive numbers, so the exponent in Eq. 10 should be $\alpha - 1$ instead of $\alpha$. The reason for adding 1 is to include a zero value. For example, when shared exponent bias is -1, an 8-bit fixed-point quantization would be expressed as follows:

$$P = \{2^0, 2^{-1}, 2^{-2}, 2^{-3}, 2^{-4}, 2^{-5}, 2^{-6}, 2^{-7}\} \tag{11}$$

$$L = \{7, 6, 5, 4, 3, 2, 1\} \tag{12}$$

The first interval from 1 to 0.5 would be evenly sliced by $2^{7-1}$ datapoints, the next interval from 0.5 to 0.25 with $2^{6-1}$, etc. Various cases are shown in Fig. 7, with $P$ plotted on the x-axis and $L$ plotted on the y-axis. Since $P$ represents the range of values due to shared exponent bias that is independent of the data format, $L$ can represent all of the various data formats we consider in this paper.

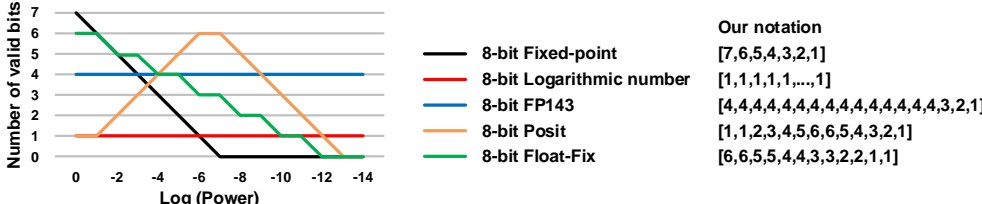

Figure 7: Number of valid bits in various 8-bit formats.

When selecting 8-bit formats, we chose the formats so that the intervals with less than 3 valid bits do not appear for more than two digits to reduce the search space, as they have an unnecessarily large dynamic range. Thus, formats such as [7,6,5,4,2,2,2,1] were excluded from the search space. Considering all of the generation rules, we selected 166 distinct 8-bit formats with different dynamic range and valid bits from [7,6,5,4,3,2,1] to [3,3,3,...,3,2,1]. After the number of valid bits for an 8-bit format is selected, 1 or 2 is subtracted from each value to create a corresponding 7-bit and 6-bit formats. For example, in the case of [6,5,5,5,5,4,4,4,3,2,1] 8-bit format, the 7-bit corresponding format is [5,4,4,4,4,3,3,3,2,1] and 6-bit corresponding format is [4,3,3,3,3,2,2,2,1]. From the generated 166 8-bit formats, 7-bit and 6-bit formats were also generated using this rule.

## A.2 SOFTWARE IMPLEMENTATION DETAILS

To support quantized training for various formats, custom C++ and CUDA codes to emulate quantized data were written. Our custom C++ and CUDA extension code could perform quantization-related functions through utilizing the Python APIs in PyTorch for extendable research while maintaining high performance. We emulate the quantized value using custom code in the part that needs quantization throughout the network, and PyTorch built-in functions are used for computation kernels such as convolution and matrix multiplication. We created a package named lptorch, short for low precision PyTorch, and the code can be found in the supplementary material.

## A.3 ANGLE VS. MAGNITUDE TO PREDICT PERFORMANCE

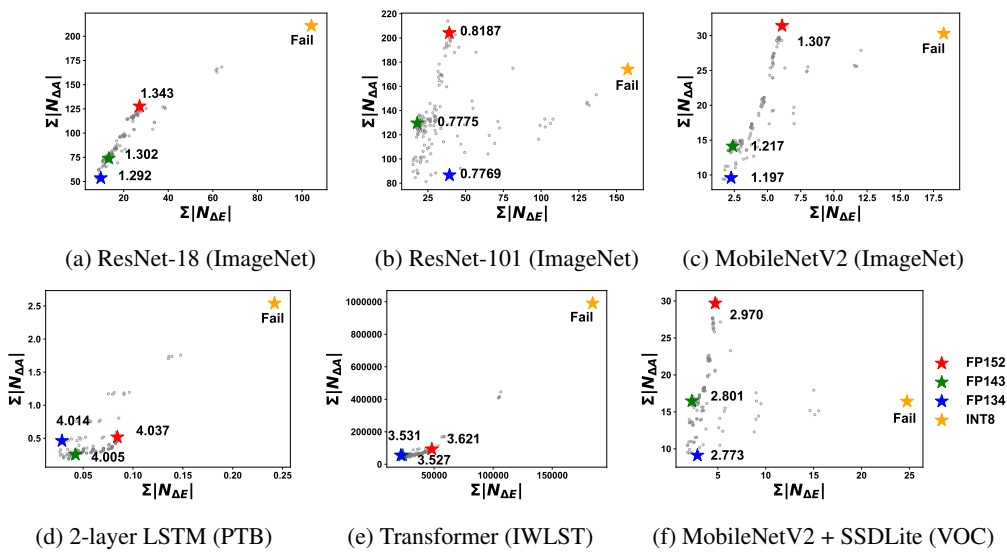

Figure 8: Scatter plots showing $|N_{\Delta E}|$ and $|N_{\Delta A}|$ of 166 8-bit quantization formats. The number next to the star represents the loss obtained through actual training.

In addition to the misalignment angles of $G_w$ ($\angle(G_w, G_w + N_{\Delta A})$ and $\angle(G_w, G_w + N_{\Delta E})$), as defined in Section 2.3, we used the magnitude of noise ($|N_{\Delta A}|$ and $|N_{\Delta E}|$) in order to predict the final trained performance, and the results are shown in Fig. 8. Fig. 3 and Fig. 8 show that both the error magnitude and the misalignment angle are good metrics for determining optimal data format. For the six target models, both metrics suggest FP134 as the best format. However, the misalignment angle still better captures the training performance. For instance, in Fig. 8(e), although FP134 shows smaller noise magnitude, the actual training loss is smaller for FP143. Similarly, in Fig. 8(b), (c) and (f), although INT8 failed and FP152 succeeded in training, the absolute value of noise did not indicate a clear superior of the two formats. Based on these observations, we conclude that the misalignment angles are more suitable for predicting training performance compared against using the absolute value of noise.

## A.4 HYSTERESIS QUANTIZATION WITH INTEGER WEIGHTS

In addition to 4-bit logarithmic weights, we also tested the hysteresis quantization scheme on a low-precision integer format (INT4) that uses uniform quantization. The results are shown in Table 5. Experimental results show that using hysteresis improves the performance in most cases. In addition, in MobileNetV2 training with INT4 weights, training initially failed, but using hysteresis enables reliable training, which suggests that hysteresis quantization not only helps the network to reach the optimal point but also prevents divergence in an unwanted direction during the training process.

However, it is interesting to see that the hysteresis quantization is less effective on the LSTM model for the INT4 format. We suspect that this is due to the weight distribution characteristics of the LSTM model. As shown in Fig. 9, most of the weights have a relatively large magnitude in the

Table 5: 4-bit Uniform Weight Training with and without Hysteresis Quantization.

| Model (Dataset) [Metric] | FP32 | INT4 W | INT4 W + Hysteresis | Δ |
|---|---|---|---|---|
| ResNet-18 (ImageNet) | 69.8 | 61.8 | **67.0** | +5.2 |
| MobileNetV2 (ImageNet) | 72.2 | Fail | **69.7** | - |
| 2-layer LSTM (PTB) [ppl.] | 91.5 | **113.4** | 118.3 | +4.9[1] |
| Transformer (IWSLT) [BLEU] | 34.8 | 32.3 | **32.4** | +0.1 |
| MobileNetV2 + SSDLite (VOC) [mAP] | 68.3 | Fail | **65.7** | - |

[1] Lower is better

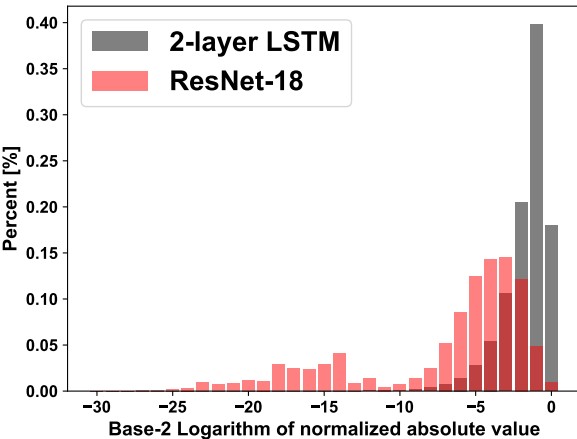

Figure 9: Weight distribution of first layer of 2-layer LSTM and ResNet-18.

LSTM model when normalized, contrary to ResNet-18 in which the weights are more evenly distributed. In logarithmic formats, the relative amount of quantization error is similar for all values. In contrast, the relative amount of quantization error is smaller for large values in uniform quantization. Therefore, the weight parameters of LSTM are more severely affected by fluctuation in logarithmic formats, making our hysteresis quantization scheme more effective in those formats compared to uniform quantization.

## A.5 EXPERIMENTAL DETAILS

### A.5.1 RESNET-18 (IMAGENET)

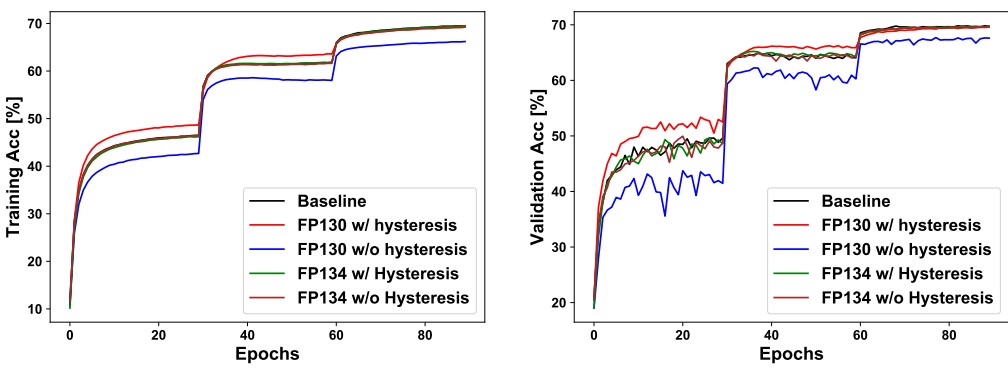

Figure 10: Top-1 Accuracy on ImageNet using a ResNet-18 model.

We conducted ImageNet experiments using SGD with a momentum of 0.9 for 90 epochs with a batch size of 256 images and an initial learning rate of 0.1 which is decayed by a factor of 10 at the 30th and 60th epochs. We used the ResNet-18 architecture from the official PyTorch implementation[1]. Fig. 10 shows Top-1 training & validation accuracy graphs. Observation of the training graph indicates that all of the results are close to the baseline within 0.2% with the exception of FP130 without hysteresis quantization.

### A.5.2 RESNET-101 (IMAGENET)

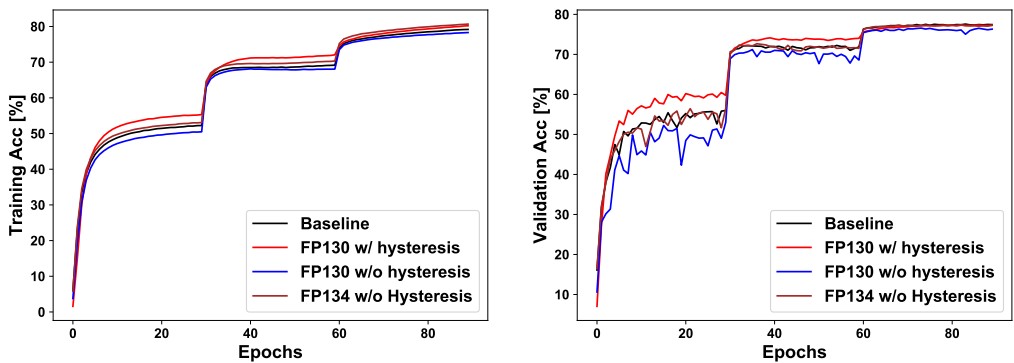

Figure 11: Top-1 Accuracy on ImageNet using a ResNet-101 model.

We trained ResNet-101 by applying the same training method as ResNet-18. We conducted ImageNet experiments using SGD with a momentum of 0.9 for 90 epochs with a batch size of 256 images and an initial learning rate of 0.1 which is decayed by a factor of 10 at the 30th and 60th epochs. We used the ResNet-101 architecture from the official PyTorch implementation[2]. Fig. 11 shows Top-1 training & validation accuracy graphs. Observation of the training graph indicates that all of the results are close to the baseline with less than 0.3% performance drop except for FP130 without hysteresis quantization.

### A.5.3 MOBILENETV2 (IMAGENET)

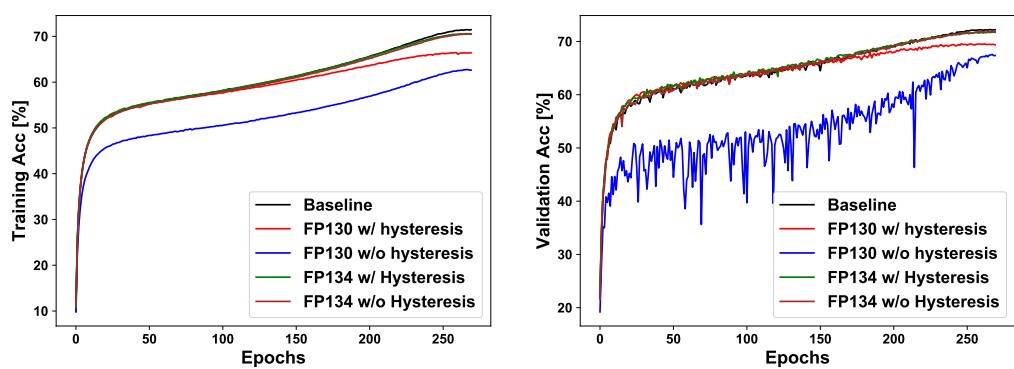

Figure 12: Top-1 Accuracy on ImageNet using a MobileNet-V2 model.

We conducted ImageNet experiments using SGD with a momentum of 0.9 for 270 epochs with a batch size of 256 images and cosine annealing with an initial learning rate of 0.05. We used the MobileNetV2 architecture from the official PyTorch implementation[3]. Fig. 12 shows Top-1 training & validation accuracy graphs. Observation of the training graph indicates that FP130

---

[1]https://github.com/pytorch/examples/tree/master/imagenet

[2]https://github.com/pytorch/examples/tree/master/imagenet

[3]https://github.com/pytorch/examples/tree/master/imagenet

without hysteresis leads to very unstable fluctuations throughout the training. On the other hand, in FP130 with hysteresis, training is less susceptible to fluctuations and follows the baseline (FP32) training closely until the learning rate decreases toward the latter part of learning, where both FP130 with hysteresis and FP134 show some degradation from the baseline. This is seen as a limitation due to the low precision of each format.

### A.5.4   2-LAYER LSTM (PTB)

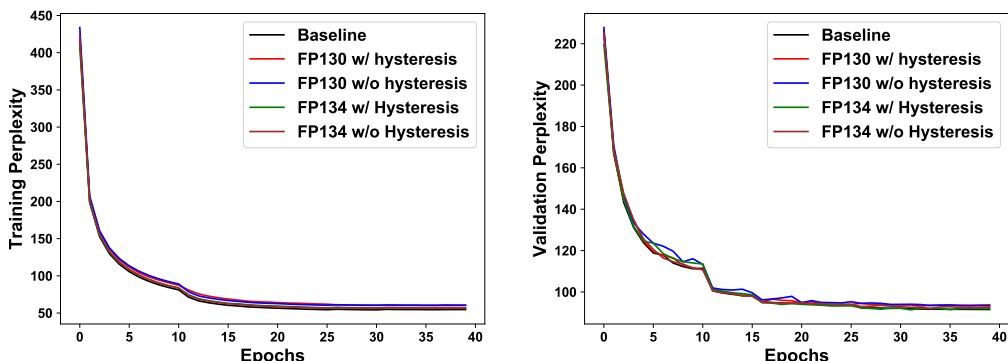

Figure 13: Perplexify on PTB using a 2-layer LSTM model.

We adopted the 2-layer Long Short Term Memory (LSTM) network from PyTorch Examples[4] for language modeling on the Penn Treebank dataset (Marcus et al., 1993). We ran experiments in batches of 20 sentences with an initial learning rate of 20 which is decayed by a factor of 4 at epoch 11, 16, 26, 31 and 37. The embedding and hidden dimensions are 650 and the sequence length is 35. Fig. 13 shows training & validation perplexity.

### A.5.5   TRANSFORMER MODEL (IWLST)

We adopted the Transformer Base model from the FairSeq[5] repository on the IWSLT'14 German to English translation task. We used Adam optimizer and default training parameters found in the repository and trained from scratch for 25 epochs. BLEU scores were calculated using the script from the repository.

### A.5.6   MOBILENETV2 + SSDLITE (VOC)

We adopted a PyTorch implementation of SSDLite from the online repository[6]. The base network is MobileNetV2 which was pretrained with each format in Appendix A.5.3. The entire network is trained on VOC2012 and VOC2007 trainval datasets and evaluated on VOC2007 validation dataset. We used SGD with a momentum of 0.9 for 200 epochs in batches of 32 images and cosine annealing with an initial learning rate of 0.01. Fig. 14 shows validation loss at every 5 epochs. Even in this experiment, in the case of FP130 without hysteresis the loss fluctuates significantly, whereas in FP130 with hysteresis learning proceeds much more stably. FP134 showed similar results to the baseline regardless of hysteresis quantization.

### A.6   MODEL QUANTIZATION METHODS

We quantized GEMM input and batchnorm input in all quantized training experiments. Among the six models used in the experiment, the quantization details for three representative structures are shown in the Fig. 15. In each structure of figure, inputs such as x, c, h, V, K, and Q are also all quantized in 8 bits.

---

[4]https://github.com/pytorch/examples/tree/master/word_language_model
[5]https://github.com/pytorch/fairseq
[6]https://github.com/qfgaohao/pytorch-ssd

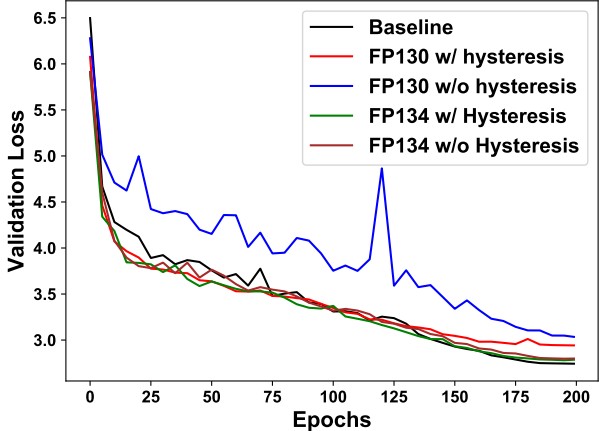

Figure 14: Validation loss on VOC using a MobileNetV2 + SSDLite.

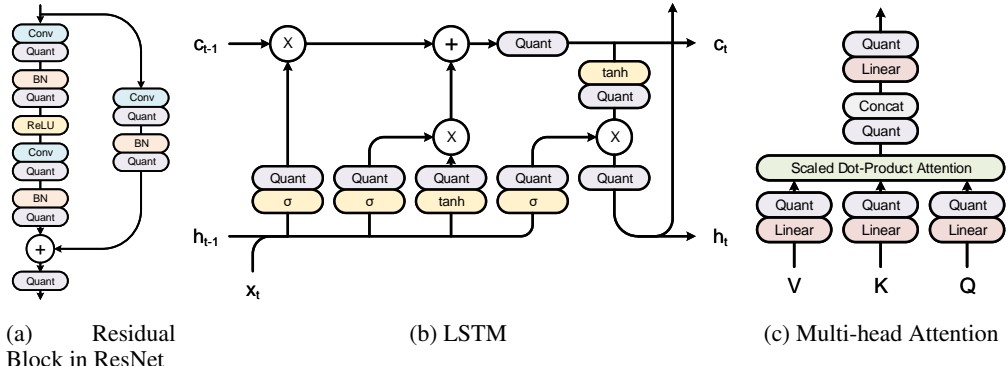

(a)    Residual          (b) LSTM          (c) Multi-head Attention
Block in ResNet

Figure 15: Quantized network structures.

## A.7   HARDWARE EVALUATION

For hardware implementation cost comparisons, we implemented a conventional MAC unit and a multi-way MAC unit with integer-based accumulation (Tambe et al., 2020; Park et al., 2021) that support data formats presented in Section 4.2. For accumulation, we use FP169 with chunk-based accumulation (Wang et al., 2018). Experimental results in Table 6 show that FP134 exhibits lower

Table 6: Area and Power of 2-Stage Pipelined MAC Units Synthesized in 40nm Process.

| MAC | Area per MAC [$\mu m^2$] | | | | | Power per MAC [$\mu W$] | | | | |
|---|---|---|---|---|---|---|---|---|---|---|
| Structure | FP134 | FP143[1] | HFP8[2] | BM8[3] | Flex16+5[4] | FP134 | FP143[1] | HFP8[2] | BM8[3] | Flex16+5[4] |
| Conventional | 1355 | 1320 | 1308 | 1460 | 3800 | 122 | 116 | 106 | 141 | 537 |
| Multi-way | | | | | | | | | | |
| 2-input | 1335 | 1480 | 2342 | 1865 | 2268 | 178 | 178 | 283 | 258 | 371 |
| 4-input | 888 | 1034 | 1615 | 1296 | 1885 | 120 | 135 | 205 | 184 | 351 |
| 8-input | 678 | 836 | 1343 | 1074 | 1672 | 97 | 123 | 194 | 168 | 342 |
| 16-input | 571 | 698 | 1065 | 957 | 1540 | 95 | 114 | 170 | 155 | 329 |
| 32-input | 511 | 668 | 994 | 898 | 1485 | 87 | 111 | 170 | 152 | 326 |
| 64-input | 509 | 638 | 955 | 856 | 1450 | 88 | 110 | 172 | 149 | 326 |

[1] Park et al. (2021) [2] Sun et al. (2019) [3] Fox et al. (2020) [4] Köster et al. (2017)

cost than FP143 and other formats in previous studies. Note that HFP8 (Sun et al., 2019) and BM8 (Fox et al., 2020) employ different formats for activation and error. Therefore, they need to be implemented in FP153 and FP145 to support all operations with a single MAC unit (Agrawal et al., 2021). Since Flex16+5 (Köster et al., 2017) requires 16-bit multiplication, its cost is significantly higher than other 8-bit formats.

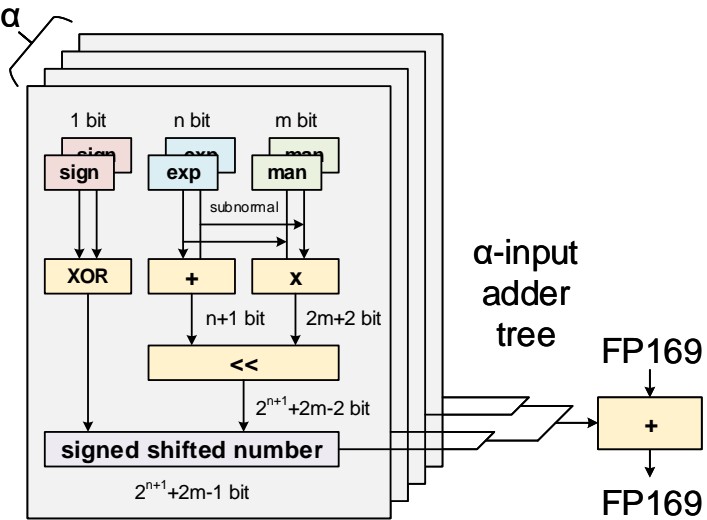

Figure 16: Multi-way MAC unit with an adder tree.

A conventional MAC unit consists of a multiplier and an accumulator. In the multiplier, the exponents of two input operands are summed while their mantissas are multiplied. The multiplication part is more complex, and hence it dominates the area of the multiplier. As a result, the size of the multiplier is larger when more bits are allocated to mantissa. In the accumulator, a floating-point adder adds the multiplication results to a partial sum in FP169. The adder is decomposed into a shifter that aligns the mantissa by the exponent difference, an integer adder that sums aligned mantissas, and a quantization unit that converts the result back to FP169. Since the result is re-quantized into FP169, the addition operation of aligned mantissas does not need to be lossless. FP169 format has a 10-bit mantissa including one hidden bit. We only need to accurately calculate higher 10 bits, which necessitates a 12-bit adder considering rounding. Shifting by more than 12 bits is not needed even if the result of the multiplier has a larger exponent range. Therefore, the shifter, adder, and quantization unit, which are the components of the accumulator, are not affected by the input format. There are minor differences such as an adder that calculates the difference between exponents and a shifter with a different bit width of the input, but their costs are ignorable.

Contrarily, a multi-way MAC consists of a multiplier, a shifter for alignment, an adder tree, a normalization unit, and a final accumulator. The multiplier and the final accumulator are identical to those of the conventional MAC. However, since only one normalization unit and one final accumulator are shared across multiple inputs, their implementation cost becomes insignificant for a larger number of inputs. The shifter for alignment converts the multiplier output to an integer format since the cost of integer addition is lower than that of floating-point addition. Then, the adder tree sums those integer values, and the normalization unit converts the result back to a floating-point format. The cost of the shifter for alignment, adder tree, and normalization unit is all determined by the integer bit width, and the larger the exponent range of the input operands, the larger the required bit width, as shown in Fig. 16. In FP134, FP143, and FP152, the minimum integer bit widths are 23, 37, and 67 bits, respectively. Since the bit width is sufficiently large, the cost difference of these units exceeds the cost difference of the multiplier. Therefore, the cost of a multi-way MAC increases with the number of exponent bits.

When designing a neural network training processor, some parts of the hardware (e.g., batch normalization, non-linear activation functions such as tanh and sigmoid, and softmax function) are typically implemented with higher precision to avoid performance drop. Hence, we need to consider data for-

Table 7: Area and Power of Format Converting Units Synthesized in 40nm Process.

| | Area [$\mu m^2$] | | | | | Power [$\mu W$] | | | | |
|---|---|---|---|---|---|---|---|---|---|---|
| Direction | FP134 | FP143 | HFP8[1] | BM8[2] | Flex16+5[3] | FP134 | FP143 | HFP8[1] | BM8[2] | Flex16+5[3] |
| To FP32 | 155 | 141 | 145 | 176 | 330 | 28 | 26 | 27 | 30 | 53 |
| From FP32 | 139 | 144 | 152 | 162 | 427 | 19 | 20 | 22 | 23 | 55 |

[1] Sun et al. (2019) [2] Fox et al. (2020) [3] Köster et al. (2017)

Table 8: FPGA Implementation Results of 2-Stage Pipelined MAC Units

| MAC | FP134 | | FP143[1] | | HFP8[2] | | BM8[3] | | Flex16+5[4] | |
|---|---|---|---|---|---|---|---|---|---|---|
| Structure | LUT | FF | LUT | FF | LUT | FF | LUT | FF | LUT | FF |
| Conventional | 284 | 35 | 269 | 33 | 273 | 33 | 304 | 35 | 606 | 69 |
| Multi-way | | | | | | | | | | |
| 2-input | 499 | 62 | 648 | 76 | 990 | 108 | 755 | 80 | 891 | 70 |
| 4-input | 719 | 63 | 958 | 77 | 1589 | 109 | 1193 | 81 | 1499 | 71 |
| 8-input | 1141 | 64 | 1503 | 78 | 2681 | 110 | 2013 | 82 | 2664 | 72 |
| 16-input | 1997 | 65 | 2603 | 79 | 4632 | 110 | 3516 | 83 | 4955 | 73 |
| 32-input | 3526 | 66 | 4758 | 80 | 8614 | 112 | 6591 | 84 | 9500 | 74 |
| 64-input | 6713 | 67 | 9074 | 81 | 16289 | 113 | 12724 | 85 | 18649 | 75 |

[1] Park et al. (2021) [2] Sun et al. (2019) [3] Fox et al. (2020) [4] Köster et al. (2017)

mat conversion overheads when comparing different formats. If we consider various 8-bit data formats with different representation methods, as we did in Table 6, and assume that computations other than MAC operations are implemented in full precision, the processing architecture (except MAC units) will be identical for all formats. In addition, the on/off-chip memory space, control logics, and on-chip interconnects will remain the same. The only difference would be the low-precision MAC units and the data conversion units between full-precision and low-precision formats. However, the cost of conversion between low-precision and high-precision floating-point formats is typically very low and does not vary much with the low-precision format. For low-precision to high-precision conversion, we only have to add a bias-correction term to the exponent and add 0 after the mantissa. For high-precision to low-precision conversion, we need to add a bias-correction term to the exponent, clamp the overflowed value to the maximum, and round off the mantissa. The cost is very low compared to the MAC operation, and the cost difference between different low-precision formats is negligible. We have synthesized the conversion units for different formats, and their costs are presented in Table 7. The experimental results confirm that the overhead of data format conversion is significantly lower than MAC operations. In addition, all formats except Flexpoint exhibit similar conversion costs.

In addition to the synthesis result for ASIC implementation in Table 6, we measured the hardware overhead of MAC units of different data formats on FPGA. Table 8 shows the synthesis results on Xilinx Artix-7 FPGA (XC7A100TCSG324-1). Those MAC units do not need block RAMs (BRAMs), and we used a compiler directive to avoid using DSP modules for fair comparisons. Table 8 shows a similar trend to Table 6; the cost of one MAC gradually decreases as the number of inputs increases in the multi-way MAC. Also, due to integer-based addition in the adder tree, the cost of FP134, which has the smallest dynamic range, exhibits lower costs than the other formats.

A.8    EFFECT OF QUANTIZATION NOISE ON DATA FLOW QUANTIZATION

Table 9 shows the training results when both activation and error are quantized in various data formats. If an appropriate amount of noise is introduced in the network during training, it will increase the training loss but reduce the validation loss, suggesting that the model has been improved due to the regularization effect. However, if the noise level continues to increase, the model's performance will start to degrade at some point. For instance, when MobileNetV2 is quantized in FP134, its performance is improved through the regularization effect since the training loss increases while

Table 9: Training Result Comparisons

| Model (Dataset) | | FP32 | | FP134 | | FP143 | | FP152 | |
|---|---|---|---|---|---|---|---|---|---|
| | | train | val | train | val | train | val | train | val |
| ResNet-18 | Loss | 1.286 | 1.220 | 1.292 | 1.223 | 1.302 | 1.233 | 1.343 | 1.261 |
| (ImageNet) | Accuracy | 69.50 | 69.73 | 69.46 | 69.67 | 69.19 | 69.63 | 68.27 | 68.93 |
| MobileNetV2 | Loss | 1.182 | 1.118 | 1.197 | 1.113 | 1.217 | 1.133 | 1.301 | 1.192 |
| (ImageNet) | Accuracy | 71.47 | 72.20 | 71.16 | 72.25 | 70.63 | 71.92 | 68.70 | 70.35 |
| 2-layer LSTM | Loss | 3.994 | 4.517 | 4.014 | 4.525 | 4.005 | 4.518 | 4.037 | 4.535 |
| (PTB) | ppl. | 54.27 | 91.54 | 55.39 | 92.30 | 54.89 | 91.68 | 56.63 | 93.20 |
| Transformer | Loss | 3.498 | 3.847 | 3.531 | 3.885 | 3.527 | 3.887 | 3.621 | 3.942 |
| (IWSLT) | BLEU | - | 34.75 | - | 34.77 | - | 34.6 | - | 33.79 |
| MobileNetV2 | Loss | - | 2.743 | - | 2.773 | - | 2.801 | - | 2.970 |
| +SSDLite(VOC) | mAP | - | 68.34 | - | 68.16 | - | 68.32 | - | 66.56 |

the validation loss decreases compared to FP32. However, both the training and validation losses increase when quantized in most cases, resulting in lower accuracy. This suggests that using a very low precision data format already introduces a large amount of noise in the network, incurring performance degradation. Hence, it is necessary to reduce error in the network to improve the training performance in low-precision training.

