# OpenReview forum: "Toward Efficient Low-Precision Training: Data Format Optimization and Hysteresis Quantization"
_ICLR.cc/2022/Conference — ICLR 2022 Poster_

### Official Review · Reviewer_mUcf · 2021-10-19

**Correctness:** 3
**Technical Novelty And Significance:** 3
**Empirical Novelty And Significance:** 3
**Recommendation:** 5
**Confidence:** 4

**Main Review:**

pros.
1. This paper addresses a very relevant topic. Though model compression has been widely discussed in the recent literature, efficient training with quantized networks is an approach worth exploring.
2. The technique part of this paper is simple and easy to follow. It is clear that the Hysteresis setting contributes to a consistent improvement over the standard approach on both image recognition and NLP tasks.

cons.
1. There are some related works missing in comparisons such as Flexpoint [1] and HFINT [2]. Especially, HFINT first notices the difference in weights distribution between CNN and NLP models/tasks then presents an adaptive floating-point format with hardware implementations. Compared with the static FP143/FP134 mode used in the current draft, they use an on-the-fly adaptive format. In light of this, it would be better to include a HARDWARE EVALUATION section to verify the effectiveness of the proposed "data flow quantization".
2. Please denote "weight gradients" as $G_{W}$ or $G^{W}$. $WG$ commonly refers to matrix multiplication.
3. Figure 2: How to distinguish 8-,7-,6-bit quantization formats?
4. It seems that Figure 8 (in A.3) has almost the same tendency as Figure 2. In general, F134 performs the best. Why introduce a more complicated metric to determine the optimal format? Note that the authors conduct all experiments under the setting of FP134.
4. How to obtain $WG+N_{\Delta E}$ and $WG+N_{\Delta A}$?
5. Table1: Why does "1-input" consume a much larger area than conventional MAC? Besides, why conventional MAC consumes a smaller area along with the decrease of mantissa, yet Multi-way MAC behaves in the opposite way? I expect more results on LUTs, DSPs, BRAMs, and Power except for Area.
6. Is hysteresis quantization compatible with uniform quantization?
7. Does FP134 still involve a scaling factor $s$ during training?

ref.
* [1] Flexpoint: An Adaptive Numerical Format for Efficient Training of Deep Neural Networks. NIPS2017
* [2] Algorithm-Hardware Co-Design of Adaptive Floating-Point Encodings for Resilient Deep Learning Inference. DAC2020

supplementary:
1. Please include a **README** file in the supplementary material.
2. How to understand the magic number $555555543210$ used for "data format for data flow quantization"?

**Summary Of The Paper:**

The submission starts from an interesting point that various quantized training environments may require different formats for training accurate deep neural networks. They present a metric based on the misalignment between $\frac{\partial \ell}{\partial w}+noise$ and $\frac{\partial \ell}{\partial w}$ to determine the optimal format. To mitigate the fluctuation issue caused by network quantization, they propose a hysteresis quantization scheme to avoid frequent changes of quantized points. The experiment results fully support the effectiveness of the proposed methods.

**Summary Of The Review:**

Overall, the idea of hysteresis quantization seems neat and new. However, the effectiveness of the performance indicator needs to be further clarified. The empirical success of FP130 with hysteresis seems to further weaken the motivation of finding the optimal data format.

The authors are encouraged to address the weakness. I will consider changing the final rating according to the authors' feedback.

---

> ### Author Response · Authors · 2021-11-17
> **Response to "Official Review 4"**
>
> We thank the reviewer for carefully reviewing our submission. We have addressed all questions and comments in the below section and made appropriate changes to the manuscript in this revision.
>
> $\textbf{Q1.}$ There are some related works missing in comparisons such as Flexpoint  and HFINT. Especially, HFINT first notices the difference in weights distribution between CNN and NLP models/tasks then presents an adaptive floating-point format with hardware implementations. Compared with the static FP143/FP134 mode used in the current draft, they use an on-the-fly adaptive format. In light of this, it would be better to include a HARDWARE EVALUATION section to verify the effectiveness of the proposed "data flow quantization".
>
> $\textbf{A1.}$ We thank the reviewer for bringing relevant prior works to our attention. As suggested by the reviewer, we have cited Flexpoint in the introduction section. HFINT [1] employs shared exponent biases in weights and activations, and utilizes multi-way MAC units and integer-based accumulation. This processing scheme is nearly identical to the one in [2], which we used as a test vehicle to verify the effectiveness of our data format optimization methods in the original submission. For instance, the FP134 data format in our paper would be identical to the 8-bit version of the AdaptivFloat format that employs a 3-bit exponent.
>
> In this work, we propose efficient methods to determine an optimal data format for low-precision training for the given models and tasks. In experiments, we find that FP134 and FP143 are the best formats for the five target models and show that FP134 results in lower hardware overhead if implemented using the processing scheme in [2]. Since the processing schemes in [1] and [2] are the same, the conclusion that FP134 is the optimal format and its implementation cost would remain unchanged even if we assume the hardware architecture in [1]. We have revised Section 4.2 to discuss this point clearly.
>
> To further address the reviewer’s concern, we have added a new hardware evaluation section as Section 4.2 in the revised manuscript. We compare the hardware implementation costs of our formats and other data formats assuming the datapath proposed in [1] and [2], which confirms that FP134 exhibits the lowest hardware overhead.
>
> [1] T. Tambe et al., “Algorithm-Hardware Co-Design of Adaptive Floating-Point Encodings for Resilient Deep Learning Inference,” DAC 2020.
>
> [2] J. Park et al., “A 40nm 4.81TFLOPS/W 8b Floating-Point Training Processor for Non-Sparse Neural Networks Using Shared Exponent Bias and 24-Way Fused Multiply-Add Tree,” ISSCC 2021.
>
> $\textbf{Q2.}$ Please denote "weight gradients" as $G_W $or $G^W$. WG commonly refers to matrix multiplication.
>
> $\textbf{A2.}$ We thank the reviewer for the suggestion. We have corrected the notation throughout the manuscript in this revision.
>
> $\textbf{Q3.}$ Figure 2: How to distinguish 8-,7-,6-bit quantization formats?
>
> $\textbf{A3.}$ We apologize for the confusion. We have updated Fig. 2 and its caption to clearly display the bit precision of each format in the revised manuscript.
>
> $\textbf{Q4.}$ It seems that Figure 8 (in A.3) has almost the same tendency as Figure 2. In general, F134 performs the best. Why introduce a more complicated metric to determine the optimal format?
>
> $\textbf{A4.}$ We thank the reviewer for pointing this out. The reviewer is certainly correct that the magnitude of the error and the misalignment angle are both good metrics for determining a data format for low-precision training. For instance, if we were to choose a data format for training five target models used in our paper, both metrics would suggest FP134 as the best format. However, the misalignment angle still captures the expected performance more accurately in some cases. For instance, if we need to choose a data format for training the transformer model, the scatter plot for error magnitude (Fig. 7(e)) suggests using FP134, whereas we could select the best format (FP143) using the misalignment angle (Fig. 3(e)). In addition, we can expect that INT8 is the worst format for all five models in Fig. 3, but this trend is less clear for the error magnitude especially in Fig. 7(f).
>
> In addition, while obtaining the misalignment angle does require additional computations compared to the error magnitude, its overhead is negligible since the part that requires the most time and computation is to obtain $G_w$, $G_w+N_{ΔE}$, and $G_w+N_{ΔA}$, which is still significantly lower than actual training. Therefore, we believe that using an accurate metric is more important in determining optimal data formats. These points are now discussed in Section 2.3 and 2.4, and Appendix A.3 in the revised manuscript.

---

> > ### Comment · Reviewer_mUcf · 2021-11-23
> > **Thanks for the detailed feedback**
> >
> > I appreciate the detailed explanation and additional experiments (as required by other reviewers). It has been such a long journey to get here, after reading over the comments from other reviewers and feedbacks.
> >
> > Unfortunately, I still have the following concerns:
> > - The proposed customized hardware design seems to not strongly relate to the optimal data format. According to "For instance, the FP134 data format in our paper would be identical to the 8-bit version of the AdaptivFloat format that employs a 3-bit exponent.", the authors may indicate that the main contribution of this work is to "determine an optimal data format for low-precision training for the given models and tasks" instead of the hardware-algorithm co-design. In light of this, I'm still not fully convinced that it is necessary to conduct a misalignment angle analysis. Note that it is quite intuitive to utilize the magnitude of the error and a "good metric for determining a data format for low-precision training." Besides, we may train FP134 and FP143 from scratch to determine the optimal format.
> > - I'm not sure whether ICLR was the best venue for the current draft. Though the idea of hysteresis quantization seems neat and new, hysteresis quantization itself seems not strong enough to be accepted by a top conference like ICLR. The authors may further explain why this paper should be submitted to a machine learning conference instead of DAC, DATE, FPGA, etc.

---

> > > ### Author Response · Authors · 2021-11-23
> > > **We greatly appreciate the reviewer’s effort in reviewing our paper**
> > >
> > > $\textbf{Q3.}$ I'm not sure whether ICLR was the best venue for the current draft. Though the idea of hysteresis quantization seems neat and new, hysteresis quantization itself seems not strong enough to be accepted by a top conference like ICLR. The authors may further explain why this paper should be submitted to a machine learning conference instead of DAC, DATE, FPGA, etc.
> > >
> > > $\textbf{A3.}$ We thank the reviewer for bringing up this issue. First of all, most of the prior studies on the data format for low-precision training we are comparing with have been presented at machine learning conferences such as ICLR, ICML, and NeurIPS [1-5]. Those studies propose various techniques to minimize the effect of quantization error during training through observing statistics of internal variables such as activation, error, and weight gradient.
> > >
> > > However, in this work, we propose a more systematic approach to low-precision training. First, we discuss that low-precision training actually consists of two different problems: data flow quantization and network quantization. In typical low-precision training, weight, activation, error, and weight gradient are all deeply quantized. However, we show that quantizing each variable affects the training performance in different ways. For data flow quantization (e.g., quantizing error and activation during training), we experimentally show that the error incurred in the weight gradient is highly correlated with the training performance and propose an efficient method to find an optimal format for those variables. On the other hand, for network quantization (e.g., quantizing weight during training), we show that weight fluctuation is closely related to the performance drop. Based on this observation, we propose a hysteresis quantization scheme to mitigate the fluctuation issue. We will clarify this point in the camera-ready version if accepted.
> > >
> > > In summary, our main contribution is neither hardware-algorithm co-design nor a specific data format suitable for low-precision training. Instead, we propose a systematic approach to low-precision training consisting of two different methods (optimal data format search and hysteresis quantization) to mitigate issues in data flow quantization and network quantization, respectively. Hence, we believe that these methods could be applied to low-precision training on a wide range of models and tasks, and a machine learning conference is the best venue for this work.
> > >
> > > [1] Yang et al., “Swalp: Stochastic weight averaging in low precision training,” ICML 2019
> > >
> > > [2] Cambier et al., “Shifted and squeezed 8-bit floating point format for low-precision training of deep neural networks,” ICLR 2020
> > >
> > > [3] Sun et al., “Hybrid 8-bit floating point (hfp8) training and inference for deep neural networks, “ NeurIPS 2019
> > >
> > > [4] Fox et al,. “A block minifloat representation for training deep neural networks,” ICLR 2020
> > >
> > > [5] K ̈oster et al., “Flexpoint:  An adaptive numerical format for efficient training of deep neural networks,” NeurIPS 2017.

---

> > > ### Author Response · Authors · 2021-11-23
> > > **We greatly appreciate the reviewer’s effort in reviewing our paper**
> > >
> > > We greatly appreciate the reviewer’s effort in reviewing our paper and response to the reviewers’ comments. We have addressed the comments and questions below.
> > >
> > > $\textbf{Q1.}$ The proposed customized hardware design seems to not strongly relate to the optimal data format. According to "For instance, the FP134 data format in our paper would be identical to the 8-bit version of the AdaptivFloat format that employs a 3-bit exponent.", the authors may indicate that the main contribution of this work is to "determine an optimal data format for low-precision training for the given models and tasks" instead of the hardware-algorithm co-design.
> > >
> > > $\textbf{A1.}$ We entirely agree with the reviewer that our main contribution is the method to determine optimal formats for low-precision training, not its hardware implementation. This is exactly what we are proposing, but it seems that the original submission was misleading and has caused confusion. In our work, we are not proposing any new hardware implementation technique. The multi-way MAC unit and other hardware designs are borrowed from prior studies on low-precision training [1], and they are only used for comparisons with other data formats. In the original submission, we wanted to rigorously show that our comparison method is technically correct and fair considering hardware implementation. However, we suspect that focusing too much on the hardware details has caused misinterpretations about our main contributions. To address this, we have moved the hardware implementation section to the appendix in the revised manuscript. We will further clarify this point and clearly describe our contributions in the camera-ready version if accepted.
> > >
> > > [1] J. Park et al., “A 40nm 4.81TFLOPS/W 8b Floating-Point Training Processor for Non-Sparse Neural Networks Using Shared Exponent Bias and 24-Way Fused Multiply-Add Tree,” ISSCC 2021.
> > >
> > >
> > > $\textbf{Q2.}$ In light of this, I'm still not fully convinced that it is necessary to conduct a misalignment angle analysis. Note that it is quite intuitive to utilize the magnitude of the error and a "good metric for determining a data format for low-precision training." Besides, we may train FP134 and FP143 from scratch to determine the optimal format.
> > >
> > > $\textbf{A2.}$ This is a very valid point. As pointed out by the reviewer, the magnitude of the error is already a good enough metric to estimate the training performance of a data format. Our main contribution in this work is that we showed that the error incurred in the weight gradient could be used to predict training performance. In this context, we suggested two different metrics (error magnitude and misalignment angle) and compared their prediction performance. We agree that those two metrics exhibit very similar prediction accuracy in most cases. To address the reviewer’s concern, we will tone down our claim that the misalignment angle is a better metric in the camera-ready version if accepted. More specifically, we will clearly show that our main contribution is the method itself (i.e., observing errors in weight gradients), not a specific metric. We will discuss that the two metrics are equally good, and the error magnitude is advantageous as it is a simple and intuitive approach, whereas the misalignment angle could predict the training performance more accurately in some cases.
> > >
> > > As pointed out by the reviewer, we could actually train the models using FP134 and FP143 to determine which one is a better format for training. However, please note that there are numerous possible data formats that could be constructed using 8 bits. As shown in Fig. 2, our method allows for quickly comparing the training performance of many different data formats. As a result, we can efficiently find promising data formats from a large set of candidates. After narrowing down to several data formats through our method, we may actually train the target models using those formats to find the best format, as suggested by the reviewer. We will also discuss this point in the camera-ready version.

---

> ### Author Response · Authors · 2021-11-17
> **Response to "Official Review 4"**
>
> $\textbf{Q5.}$ How to obtain $WG+N_{ΔE}$ and $WG+N_{ΔA}$?
>
> $\textbf{A5.}$ $N_{ΔE}$ and $N_{ΔA}$ are the noise introduced in the weight gradient when the error is quantized and the activation is quantized, respectively. Therefore, if we quantize the error during training, then the obtained weight gradient is $G_W+N_{ΔE}$. Similarly, if the activation is quantized during training, the resulting weight gradient is $G_W+N_{ΔA}$. In our experiments, we measured those variables on the models that are pre-trained in full precision. While this requires one-time training in full precision, we do not need additional training for each data format candidate to predict their training performance. In addition, we observe these values for the weights of the first layer to minimize implementation overhead since they contain the noise in activations and errors of all layers. We convert the weight gradient into a one-dimensional vector and obtain cosine similarity between vectors (misalignment angle) or its L1-norm (error magnitude). This information has been added to Section 2.3.
>
> $\textbf{Q6.}$ Table1: Why does "1-input" consume a much larger area than conventional MAC? Besides, why conventional MAC consumes a smaller area along with the decrease of mantissa, yet Multi-way MAC behaves in the opposite way? I expect more results on LUTs, DSPs, BRAMs, and Power except for Area.
>
> $\textbf{A6.}$ We thank the reviewer for bringing up an important point. We apologize that this aspect was not discussed in detail in our original submission. A conventional MAC unit consists of a multiplier and an accumulator. In the multiplier, the exponents of two input operands are summed while their mantissas are multiplied. The multiplication part is more complex, and hence it dominates the area of the multiplier. As a result, the size of the multiplier is larger when more bits are allocated to mantissa. In the accumulator, a floating-point adder adds the multiplication results to a partial sum in FP169. The adder is decomposed into a shifter that aligns the mantissa by the exponent difference, an integer adder that sums aligned mantissas, and a quantization unit that converts the result back to FP169. Since the result is re-quantized into FP169, the addition operation of aligned mantissas does not need to be lossless. FP169 format has a 10-bit mantissa including one hidden bit. Therefore, we only need to accurately calculate higher 10 bits, which necessitates a 12-bit adder considering rounding. Therefore, shifting by more than 12 bits is not needed even if the result of the multiplier has a larger exponent range. Therefore, the shifter, adder, and quantization unit, which are the components of the accumulator, are not affected by the input format. There are minor differences such as an adder that calculates the difference between exponents and a shifter with a different bit width of the input, but their costs are ignorable.
>
> Contrarily, a multi-way MAC consists of a multiplier, a shifter for alignment, an adder tree, a normalization unit, and a final accumulator. The multiplier and the final accumulator are identical to those of the conventional MAC. However, since only one normalization unit and one final accumulator are shared across multiple inputs, their implementation cost becomes insignificant for a larger number of inputs. The shifter for alignment converts the multiplier output to an integer format since the cost of integer addition is lower than that of floating-point addition. Then, the adder tree sums those integer values, and the normalization unit converts the result back to a floating-point format. The cost of the shifter for alignment, adder tree, and normalization unit is all determined by the integer bit width, and the larger the exponent range of the input operands, the larger the required bit width, as shown in Fig. 15. In FP134, FP143, and FP152, the minimum integer bit widths are 23, 37, and 67 bits, respectively. Since the bit width is sufficiently large, the cost difference of these units exceeds the cost difference of the multiplier. Therefore, the cost of a multi-way MAC increases with the number of exponent bits.
>
> Please note that the 1-input multi-way MAC shows a much higher cost than the conventional MAC due to an unnecessary shifter for alignment and normalization unit. However, we agree that this information is confusing, so we have removed the 1-input MAC from Table 1 (now Table 3) and added discussions to Section 4.2 and Appendix A.7.
>
> In the original submission, we synthesized MAC units using standard cells in 40nm CMOS process for ASIC implementation. However, following the reviewer’s suggestion, we additionally synthesized the designs on FPGA, and the experimental results are now provided in Appendix A.7. In addition, the power consumption of each design has been included in Table 3.

---

> ### Author Response · Authors · 2021-11-17
> **Response to "Official Review 4"**
>
> $\textbf{Q7.}$ Is hysteresis quantization compatible with uniform quantization?
>
> $\textbf{A7.}$ Yes, hysteresis quantization is applicable to any quantization scheme as long as it is deterministic and the quantization points are fixed. This point has been clarified in Section 3.3, and we have added experimental results on INT4 to Appendix A.4.
>
> $\textbf{Q8.}$ Does FP134 still involve a scaling factor s during training?
>
> $\textbf{A8.}$ Yes. As we discussed in answer to Q1 above, we employ shared exponent biases [1-2] in the datapath. The shared exponent biases are effectively identical to the scaling factor. For instance, if a variable has a value of $m\*2^e$ and a shared exponent bias of $bias$, then its actual value is $m\*2^{e+bias}$, which is identical to the scaling factor of $2^{bias}$. This point has been clarified in Section 1.
>
> [1] T. Tambe et al., “Algorithm-Hardware Co-Design of Adaptive Floating-Point Encodings for Resilient Deep Learning Inference,” DAC 2020.
>
> [2] J. Park et al., “A 40nm 4.81TFLOPS/W 8b Floating-Point Training Processor for Non-Sparse Neural Networks Using Shared Exponent Bias and 24-Way Fused Multiply-Add Tree,” ISSCC 2021.
>
> $\textbf{Q9.}$ The empirical success of FP130 with hysteresis seems to further weaken the motivation of finding the optimal data format.
>
> $\textbf{A9.}$ We thank the reviewer for pointing out an important point. In Section 2, we analyze the effect of error and activation quantizations and propose a method to find an optimal number format that minimizes those effects. The effect of weight quantization is not considered in this method. However, weights should also be deeply quantized to reduce hardware overhead in low-precision training. Therefore, we propose a hysteresis quantization scheme that mitigates performance drop due to weight quantization, making it supplementary to optimal data format selection. This point is now discussed in Section 5.
>
>
> Supplementary
>
> $\textbf{Q1.}$ Please include a README file in the supplementary material.
>
> $\textbf{A1.}$ We thank the reviewer for catching this. A README file has been added to the supplementary material. A brief description of the codes and instructions are included in the file.
>
> $\textbf{Q2.}$ How to understand the magic number 555555543210 used for "data format for data flow quantization"?
>
> $\textbf{A2.}$ We apologize for the confusion. This number references the data format introduced in Appendix A.1. This number represents the number of significand bits for a particular data interval in Eq. 8. The data intervals are expressed by \{$2^{s+1}$,$2^s$,$2^{s-1}$,…,$2^{s-(K-1)}$\}, where s and K represent the shared exponent bias and the length of the number list, respectively. In the case of 555555543210, for the first 7 data intervals (a normal exponent range with exponent values 7\~1), the number of significand bits are 5, which translates to 4 bits of mantissa. The next 4 data intervals are data intervals for an exponent value of 0, or ‘subnormal’ in a floating-point format. 4321 respectively represents 8, 4, 2, 1 quantization points inside intervals of ($2^{s-6}$\~$2^{s-7}$), ($2^{s-7}$\~$2^{s-8}$), ($2^{s-8}$\~$2^{s-9}$), and ($2^{s-9}$\~$2^{s-10}$). A closer look reveals that this is equivalent to uniform quantization inside a subnormal range. 0 simply means the end of the data format. As an additional example for better understanding, we could express as 76543210 for INT8, 4444444444444443210 for FP143, and 11111110 for 4-bit logarithmic format.  While this may seem unintuitive at first look, this expression was chosen as it gives a method for systematical generation of quantization functions in our analysis. We have revised Appendix A.1 to show this point more clearly.

---

### Official Review · Reviewer_mPme · 2021-10-30

**Correctness:** 3
**Technical Novelty And Significance:** 3
**Empirical Novelty And Significance:** 3
**Recommendation:** 6
**Confidence:** 4

**Main Review:**

**Contribution:**

1.	The authors propose a metric to evaluate the performance of different numeric formats. Using the proposed metric, the authors further find an optimal 8-bit numeric format suitable for various models.

2.	The authors find that the performance degradation of 8-bit training is due to the fluctuation issue of quantized weights. To solve this, the authors propose a hysteresis quantization scheme to improve the performance of from-scratch training.

3.	Experiments on 8-bit and 4-bit training show the promising performance of the proposed method.

**Questions and points needed to be improved:**

1.	In Figure 2, the improvement of spearman’s correlation of the proposed metric over the magnitude of the error is marginal (0.9283 vs. 0.9215). It seems that the magnitude of the error is a good metric to measure the performance degradation. What are the advantages of the proposed metric? More explanations and results are required.

2.	In Section 3.1, the authors state that the amount of change in the quantized weight due to the fluctuation is not necessarily proportional to the weight gradient. To mitigate the fluctuation issue above, the authors propose hysteresis quantization scheme. However, Figure 5 can not show the effect of hysteresis quantization. The number of $Q_w$ changes is the same in Figure 5(a) and Figure 5(b). More explanations are required.

3.	The idea of changing the rounding function in network quantization is similar to AdaRound[1]. It would be better for the authors to add more discussions on the difference between the proposed method and AdaRound.

4.	In Table 4, many notations are unclear. What do “X” and “O” denote? What do dw and x denote? More explanations are required.

**Reference:**

[1] Up or Down? Adaptive Rounding for Post-Training Quantization. ICML 2020.



**Summary Of The Paper:**

The authors propose a method to predict the performance of different numeric formats, which allows determining the optimal data format for various neural network architectures, datasets, and tasks efficiently. By comparing 498 formats in total, the authors find an optimal 8-bit format suitable for various models. To improve the performance of from-scratch training, the authors further propose hysteresis quantization to mitigate the ﬂuctuation issue. Experiments on 8-bit and 4-bit training demonstrate the effectiveness of the proposed method.

**Summary Of The Review:**

Experiments show promising performance. However, there are still some concerns regarding the proposed method. Some notations in the experiments are not clear. More explanations are required.

---

> ### Author Response · Authors · 2021-11-17
> **Response to "Official Review 3"**
>
> We thank the reviewer for carefully reviewing our submission. We have addressed all questions and comments in the below section and made appropriate changes to the manuscript in this revision.
>
> $\textbf{Q1.}$ In Figure 2, the improvement of spearman’s correlation of the proposed metric over the magnitude of the error is marginal (0.9283 vs. 0.9215). It seems that the magnitude of the error is a good metric to measure the performance degradation. What are the advantages of the proposed metric? More explanations and results are required.
>
> $\textbf{A1.}$ We thank the reviewer for this valuable feedback. In this work, we propose that observing the errors on the weight gradients is an effective method to predict the training performance of different data formats. In this context, we evaluate two metrics: error magnitude and misalignment angle. We agree with the reviewer that these two metrics might be similarly good for determining a data format for low-precision training. For instance, if we were to choose a data format for training five target models used in our paper, both metrics would suggest FP134 as the best format. However, the misalignment angle still captures the expected performance more accurately in some cases. For instance, if we were to choose a data format for training the transformer model, the scatter plot for error magnitude (Fig. 7(e)) suggests using FP134, whereas we could select the best format (FP143) using the misalignment angle (Fig. 3(e)). Similarly, we can expect that INT8 is the worst format for all five models in Fig. 3, but this trend is less clear for the error magnitude especially in Fig. 7(f).
>
> In addition, while these two metrics exhibit similar Spearman’s correlation, the shape of the loss-error curve is more distinct when using the misalignment angle for activation quantization (Fig. 2(b)). Using the misalignment angle shows a clearer trend between the loss and the error especially when the training loss is low, which is the area of our interest since we would like to choose a data format with a training loss as low as that of full-precision training. We now discuss this point in Section 2.3 and 2.4, and Appendix A.3 in the revised manuscript.
>
> $\textbf{Q2.}$ Figure 5 can not show the effect of hysteresis quantization. The number of Qw changes is the same in Figure 5(a) and Figure 5(b). More explanations are required.
>
> $\textbf{A2.}$ We apologize for the confusion. We have updated Fig. 5 (now Fig. 4) to clearly show the difference. In conventional quantization, the amount of change of the quantized weight Qw is identical for small and large weight changes (ΔW). Contrarily, in hysteresis quantization, if the weight change is small, then enough number of those changes should be accumulated to flip Qw. Hence, the update frequency is now proportional to the weight gradient. This point has been clarified in Section 3.1 and 3.2.
>
> $\textbf{Q3.}$ It would be better for the authors to add more discussions on the difference between the proposed method and AdaRound.
>
> $\textbf{A3.}$ We thank the reviewer for the suggestion. Nagel et al. (2020) showed that conventional quantization (round to nearest) might not be the optimal quantization method and proposed a new method, AdaRound. This scheme learns whether each weight should be rounded up or down to produce the same output as high-precision weights. While AdaRound was developed for post-training quantization, it might also be adopted in low-precision training to mitigate the quantization error in the weights. However, whenever full-precision weights are updated (e.g., each minibatch), we need to re-train the learnable parameters (i.e., quantization scheme of each weight), which incurs a large overhead. This discussion has been added to Section 3.2 in the revised manuscript.
>
> $\textbf{Q4.}$ In Table 4, many notations are unclear. What do “X” and “O” denote? What do dw and x denote? More explanations are required.
>
> $\textbf{A4.}$ We apologize that our notations were not very clear in the original manuscript. In the “From Scratch” column, “X” represents that the model was not trained from scratch but fine-tuned from a pre-trained model, whereas “O” represents that the model was trained from scratch with quantized variables. The variables w, x, dw, and dx represent weight, activation, weight gradient, and error (activation gradient), respectively. This information has been added to Table 4 and Section 4.1 in this revision.

---

> > ### Comment · Reviewer_mPme · 2021-11-23
> > **Feedback on rebuttal**
> >
> > I have read all the comments and thank the authors for their replies. The authors have addressed my concerns. Therefore, I tend to keep my score.

---

### Official Review · Reviewer_i9MY · 2021-10-31

**Correctness:** 2
**Technical Novelty And Significance:** 2
**Empirical Novelty And Significance:** 2
**Recommendation:** 5
**Confidence:** 5

**Main Review:**

Introduction discusses the difference between quantized models (after fine-tuning a pre-trained model) and from-scratch training for quantization (to gain speed-up for training) clearly and the direction of research to investigate why from-scratch training methods show increased degradation in accuracy is certainly important.

Unfortunately, this reviewer finds the following serious concerns in this paper.

(1) The authors argue that measuring the misalignment angle is better than the magnitude of the error. But in Section 2.3, it is not clear why the noise in the opposite direction is harmful. Similar to stochastic variation, noise on gradients can show some benefits if the amount of error is right such that regularization effects can be obtained. Moreover, as shown in Figure 2, reducing the magnitude of error also tends to present the training loss change. Is it difficult or impossible to find FP134 as an optimal one with magnitude-based error measurement? Supporting data to validate the claim (that the angle is better than the magnitude) needs to be provided.

(2) This paper seems to suggest a design methodology of NPU in the form of ASIC. Then, the authors need to prove that FP134 can be applied to a wide range of models. Such a particular format would show significant accuracy degradation if model size increases. What would be the limit of such a format? Moreover, the authors show the superiority of FP134 using ResNet-18 model while a few additional experimental results using some simple Transformers, such a strong argument to choose FP134 as an optimal one needs to consider what kind of limitations would be provided. For example, schemes in Table 3 might not be good for ResNet-18 but may be good for ResNet-101. This reviewer is not sure whether FP134 is a customized one for small models such as ResNet-18.

(3) A quantization technique using Hysteresis is interesting. But more detailed discussions and theories why Hysteresis is important for 4-bit Log W need to be included. Is Hysteresis generally helpful for other quantization formats and larger models as well?



**Summary Of The Paper:**

This paper proposes a method to find an optimal quantization format based on error angle estimation and hardware overhead. The authors also present an hysteresis-based quantization method to reduce fluctuation of exponent values such that training (from the scratch) using only 4-bit weights can result in negligible amount of accuracy degradation for ResNet-18 on ImageNet. Experimental results are provided for ResNet-18, MobileNetv2, 2-layer LSTM, Transformer, and MobileNetV2+SSDLite. For 8-bit quantization, FP134 is chosen and such quantization is also applied to BatchNorm layers to reduce memory consumption.

**Summary Of The Review:**

Overall, even though this reviewer finds some interesting ideas including Hysteresis and experimental results are good for ResNet-18, the followings need to be addressed.

(a) FP134 cannot be found by measuring the magnitude of the error?

(b) Do the authors suggest FP134 as a format to be applied to a wide range of models (especially even with large model size?). If so, please provide supporting data and theories.

(c) Most experimental data are only for ResNet-18.

---

> ### Author Response · Authors · 2021-11-17
> **Response to "Official Review 2"**
>
> We thank the reviewer for carefully reviewing our submission. We have addressed all questions and comments in the below section and made appropriate changes to the manuscript in this revision.
>
> $\textbf{Q1.}$ It is not clear why the noise in the opposite direction is harmful. Similar to stochastic variation, noise on gradients can show some benefits if the amount of error is right such that regularization effects can be obtained.
>
> $\textbf{A1.}$ We thank the reviewer for bringing up an important point. As pointed out by the reviewer, noise on gradients could also improve performance through the regularization effect by preventing overfitting. In the revised manuscript, we have included additional experimental results showing training performance as a new section in the appendix (Section A.8). Table 8 displays the training results when both activation and error are quantized in various formats. If an appropriate amount of noise is introduced in the network during training, it will increase the training loss but reduce the validation loss, suggesting that the model has been improved due to the regularization effect. However, if the noise level continues to increase, the model’s performance will start to degrade at some point. For instance, when MobileNetV2 is quantized in FP134, its performance is improved through the regularization effect since the training loss increases while the validation loss decreases compared to FP32. However, both the training and validation losses increase when quantized in most cases, resulting in lower accuracy. This suggests that using a very low precision data format already introduces a large amount of noise in the network, incurring performance degradation. As a result, we believe that introducing more noise in low-precision training does not improve the performance further through the regularization effect, which is in line with our experimental results. This point is now discussed in Section 2.3 and Appendix A.8 in the revised manuscript.
>
> $\textbf{Q2.}$ As shown in Figure 2, reducing the magnitude of error also tends to present the training loss change. Is it difficult or impossible to find FP134 as an optimal one with magnitude-based error measurement? Supporting data to validate the claim (that the angle is better than the magnitude) needs to be provided.
>
> $\textbf{A2.}$ We thank the reviewer for pointing this out, and we apologize that our original submission did not discuss this point in detail. The reviewer is certainly correct that the magnitude of the error and the misalignment angle are both good metrics for determining a data format for low-precision training. For instance, if we were to choose a data format for training five target models used in our paper, both metrics would suggest FP134 as the best format. However, the misalignment angle still captures the expected performance more accurately in some cases. For instance, if we need to choose a data format for training the transformer model, the scatter plot for error magnitude (Fig. 7(e)) suggests using FP134, whereas we could select the best format (FP143) using the misalignment angle (Fig. 3(e)). In addition, we can expect that INT8 is the worst format for all five models in Fig. 3, but this trend is less clear for the error magnitude especially in Fig. 7(f).
>
> In summary, we agree with the reviewer that both metrics could be used for determining data formats in our target models. However, the misalignment angle still better captures the training performance and allows for more accurate performance comparisons between different formats. We now discuss this point in Section 2.3 and 2.4, and Appendix A.3 in the revised manuscript.

---

> ### Author Response · Authors · 2021-11-17
> **Response to "Official Review 2"**
>
> $\textbf{Q3.}$ This paper seems to suggest a design methodology of NPU in the form of ASIC. Then, the authors need to prove that FP134 can be applied to a wide range of models. Such a particular format would show significant accuracy degradation if model size increases. What would be the limit of such a format? Moreover, the authors show the superiority of FP134 using ResNet-18 model while a few additional experimental results using some simple Transformers, such a strong argument to choose FP134 as an optimal one needs to consider what kind of limitations would be provided. For example, schemes in Table 3 might not be good for ResNet-18 but may be good for ResNet-101. This reviewer is not sure whether FP134 is a customized one for small models such as ResNet-18.
>
> $\textbf{A3.}$ We apologize for the confusion. In this work, what we are proposing is an efficient method to determine an optimal data format for the given models and tasks. By observing the error magnitude or the misalignment angle, we could predict the training performance of different data formats without actual training. In this context, we apply our method to find an optimal format for training five specific models that have been widely used in prior studies on low-precision training. Our method predicts that FP134 is the best format for training those five models, and the experimental results in Table 3 (now Table 2) confirm that this format results in better training performance than prior handpicked data formats.
>
> Hence, we are not claiming that FP134 is the best format for larger models such as ResNet-101. If we were to find an optimal format for training a wide range of neural networks, we need to apply our methodology to a larger set of models as we did for five models in this paper. This would necessitate more experiments, but its overhead is significantly lower than actually training all of these neural networks. We have clarified this point in Section 2.4.
>
> In addition, to validate the scalability of our methodology, we additionally estimated the error magnitude and the misalignment angle for a larger network, ResNet-101. The results have been added as Fig. 3(b) and Fig. 7(b). The scatter plot suggests that FP134 and FP143 are the best candidates as they exhibit the lowest misalignment angle for activation and error, respectively. Considering that FP134 has a lower hardware cost, we also trained ResNet-101 using the FP134 format, and the trained model only shows a 0.2% performance drop compared to FP32 training (77.6% vs. 77.4%). This result shows that our approach could be applied to larger models. Please note that the actual losses have not been obtained yet in Fig. 3(b) and Fig. 7(b). We are still training ResNet-101 in different formats to obtain these values, and we will update them once those runs are finished.

---

> > ### Comment · Reviewer_i9MY · 2021-11-18
> > **Thanks for detailed responses**
> >
> >
> > First of all, I greatly appreciate high efforts of the authors to revise the manuscript and address reviewers' concerns.
> > Unfortunately, this reviewer still has serious concerns on this manuscript. The authors mentioned in the response as the following: "In this work, what we are proposing is an efficient method to determine an optimal data format for the given models and tasks. By observing the error magnitude or the misalignment angle, we could predict the training performance of different data formats without actual training. In this context, we apply our method to find an optimal format for training five specific models that have been widely used in prior studies on low-precision training. Our method predicts that FP134 is the best format for training those five models, and the experimental results in Table 3 (now Table 2) confirm that this format results in better training performance than prior handpicked data formats."
> >
> > - The results of this manuscript are not very surprising because if a certain NPU is specifically targeted only to a limited number of models, then area or power can be improved a lot compared to other formats that are broadly targeting various models. In other words, if we limit design space exploration, then optimization process would be different from the case that does not consider only particular models. FP134 has been derived from 5 specific models (which are relatively small). This reviewer does not think that such NPU design targeting only selected models can be useful in the field when NPU may need to support even futuristic models. Design space exploration would need some margins even to run unknown models.
> >
> > - In addition, comparison with previous formats does not seem to be fair. For example, in table 3, various formats are described with corresponding power and area. The difference is not significant while the overall architectural changes can be vastly different. For example, a lot of highly non-linear activation functions would need to be implemented by full-precision formats (otherwise, designers need to come up with approximated versions of those non-linear functions to support each different low-precision format). What would be conversion cost for those various formats (or cost for specially designed non-linear functions)? Without such additional cost and entire architectural pictures, it is difficult to estimate the overall hardware design efforts.
> >
> > - Overall, this reviewer is not sure whether the proposed design methodology can be practical for designing NPUs if the ultimate goal is to target only specific models. Hence, this reviewer keeps the original score.

---

> > > ### Author Response · Authors · 2021-11-19
> > > **Thanks for continuing the discussion**
> > >
> > > $\textbf{Q2.}$ In addition, comparison with previous formats does not seem to be fair. For example, in table 3, various formats are described with corresponding power and area. The difference is not significant while the overall architectural changes can be vastly different. For example, a lot of highly non-linear activation functions would need to be implemented by full-precision formats (otherwise, designers need to come up with approximated versions of those non-linear functions to support each different low-precision format). What would be conversion cost for those various formats (or cost for specially designed non-linear functions)? Without such additional cost and entire architectural pictures, it is difficult to estimate the overall hardware design efforts.
> > >
> > > $\textbf{A2.}$ We thank the reviewer for this insightful comment. The reviewer is certainly correct that some parts of the hardware (e.g., batch normalization, non-linear activation functions such as tanh and sigmoid, and softmax function) are typically implemented with higher precision to avoid performance drop, and we need to consider data format conversion overheads when comparing different formats.
> > >
> > > If we consider various 8-bit data formats with different representation methods, as we did in Table 3, and assume that computations other than MAC operations are implemented in full precision, the processing architecture (except MAC units) will be identical for all formats. In addition, the on/off-chip memory space, control logics, and on-chip interconnects will remain the same. The only difference would be the low-precision MAC units and the data conversion units between full-precision and low-precision formats, as pointed out by the reviewer.
> > >
> > > However, the cost of conversion between low-precision and high-precision floating-point formats is typically very low and does not vary much with the low-precision format. For low-precision to high-precision conversion, we only have to add a bias-correction term to the exponent and add 0 after the mantissa. For high-precision to low-precision conversion, we need to add a bias-correction term to the exponent, clamp the overflowed value to the maximum, and round off the mantissa. The cost is very low compared to the MAC operation, and the cost difference between different low-precision formats is negligible. Therefore, it was judged that the dominant difference according to the format was the MAC unit, and previous studies also compared only the MAC operation cost [2][4].
> > >
> > > To further address the reviewer’s concern, we have synthesized the conversion units for different formats, and their costs are presented below. The experimental results confirm that the overhead of data format conversion is significantly lower than MAC operations. In addition, all formats except Flexpoint exhibit similar conversion costs. The discussions and experimental results above have been added to the revised manuscript as Appendix A.9.
> > >
> > > notation: Area [$\mu m^2$] / Power [$\mu W$]
> > >
> > > ||Conversion to FP32|Conversion from FP32|
> > > |---|:---:|:---:|
> > > |FP134|155 / 28|139 / 19|
> > > |FP143|141 / 26| 144 / 20|
> > > |HFP8 [1]|145 / 27| 152 / 22|
> > > |BM8 [2]|176 / 30|162 / 23|
> > > |Flex16+5 [3]|330 / 53|427 / 55|
> > >
> > > [1] Sun et al., “Hybrid 8-bit floating point (hfp8) training and inference for deep neural networks, “ NeurIPS 2019
> > >
> > > [2] Fox et al,. “A block minifloat representation for training deep neural networks,” ICLR 2020
> > >
> > > [3] K ̈oster et al., “Flexpoint:  An adaptive numerical format for efficient training of deep neural networks,” NeurIPS 2017.
> > >
> > > [4] T. Tambe et al., “Algorithm-Hardware Co-Design of Adaptive Floating-Point Encodings for Resilient Deep Learning Inference,” DAC 2020.

---

> > > ### Author Response · Authors · 2021-11-19
> > > **Thanks for continuing the discussion**
> > >
> > > We thank the reviewer for providing valuable feedback and continuing the discussion. We have addressed all comments and questions below and made appropriate changes to the manuscript in this revision.
> > >
> > > $\textbf{Q1.}$ The results of this manuscript are not very surprising because if a certain NPU is specifically targeted only to a limited number of models, then area or power can be improved a lot compared to other formats that are broadly targeting various models. In other words, if we limit design space exploration, then optimization process would be different from the case that does not consider only particular models. FP134 has been derived from 5 specific models (which are relatively small). This reviewer does not think that such NPU design targeting only selected models can be useful in the field when NPU may need to support even futuristic models. Design space exploration would need some margins even to run unknown models.
> > >
> > > $\textbf{A1.}$ We apologize that our response was not clear enough and caused confusion. We are not targeting only a limited number of models. We are aiming at a data format that could be used for a wide range of models, and hence we selected five (six after revision; we thank again the reviewer for suggesting ResNet-101) representative models with different neural network architectures, layer types, sizes, and target tasks for experiments. Please note that this is consistent with the experimental setups in prior studies on low-precision data formats for training various models. The data formats we used for comparisons and their test models (datasets) are as follows:
> > >
> > > SWALP [1] – VGG16 (CIFAR-10, CIFAR-100), PreResNet-164 (CIFAR-10, CIFAR-100), ResNet-18 (ImageNet)
> > >
> > > S2FP8 [2]  – ResNet-20, 34, 50 (CIFAR-10), ResNet-18, 50 (ImageNet), Transformer tiny (En-Vi), NCF (MovieLens)
> > >
> > > HFP8 [3] – AlexNet (ImageNet), ResNet-18, 50 (ImageNet), MobileNetV2 (ImageNet), DensNet121 (ImageNet), 2-layer LSTM (PTB), Transformer-based (IWLST), 4-bidirectional-LSTM speech (SWB300), MaskRCNN with ResNet50 (COCO), SSD-Lite with MobileNetV2 (VOC)
> > >
> > > BM8 [4] – AlexNet (ImageNet), ResNet-18 (ImageNet), EfficientNet-b0 (small ImageNet), 2-layer LSTM (PTB), Transformer-based (IWSLT), SSD-Lite with MobileNetV2 (VOC)
> > >
> > > Flexpoint [5] – AlexNet (ImageNet), ResNet with 110 layers (CIFAR-10), WGAN (LSUN)
> > >
> > > Ours – ResNet-18 (ImageNet), ResNet-101 (ImageNet), MobileNetV2 (ImageNet), 2-Layer LSTM (PTB), Transformer-based (IWLST), SSD-Lite with MobileNetV2 (VOC)
> > >
> > > Comparing the models covered in previous studies and our paper, we dealt with similar or more diverse and larger models than the others, except for [3]. In the case of [3], the models are slightly more diverse, but a similar range of tasks (image classification, object detection, speech, and translation) and models with similar layer types (convolution, LSTM, transformer, and fully-connected layers) and size (DensNet121 vs. ResNet-101) are tested. Hence, in the same design space, our methodology determined a format with better performance and less hardware cost. We wanted to be conservative on our claim in the previous revision, but it seems that this change caused confusion to the reviewer about our ultimate goal in this work. We apologize for the confusion again. We have updated Section 2.4 and 4.1 to clarify this point in this revision.
> > >
> > > In addition, our design methodology would be more advantageous for a larger exploration space consisting of more models. In conventional approaches, one needs to actually train each network using custom data formats to compare their performance, but our performance prediction scheme could significantly accelerate this process as it only needs observing the errors in the weight gradients without from-scratch training for each format.
> > >
> > > [1] Yang et al., “Swalp: Stochastic weight averaging in low precision training,” ICML 2019
> > >
> > > [2] Cambier et al., “Shifted and squeezed 8-bit floating point format for low-precision training of deep neural networks,” ICLR 2020
> > >
> > > [3] Sun et al., “Hybrid 8-bit floating point (hfp8) training and inference for deep neural networks, “ NeurIPS 2019
> > >
> > > [4] Fox et al,. “A block minifloat representation for training deep neural networks,” ICLR 2020
> > >
> > > [5] K ̈oster et al., “Flexpoint:  An adaptive numerical format for efficient training of deep neural networks,” NeurIPS 2017.

---

> ### Author Response · Authors · 2021-11-17
> **Response to "Official Review 2"**
>
> $\textbf{Q4.}$ More detailed discussions and theories why Hysteresis is important for 4-bit Log W need to be included. Is Hysteresis generally helpful for other quantization formats and larger models as well?
>
> $\textbf{A4.}$ We thank the reviewer for this valuable feedback. First of all, hysteresis is not particularly important for logarithmic weights. The fluctuation problem occurs not only in logarithmic formats but also in other formats employing uniform quantization (e.g., integer formats). Hence, the hysteresis quantization is applicable to other quantization formats. In the original submission, we chose the logarithmic format since it is suitable for hardware-efficient low-precision training due to its low implementation cost. This point has been clarified in Section 3.3.
>
> Also, we tested our hysteresis quantization scheme on a 4-bit integer format that relies on uniform quantization, and the experimental results have been added to Appendix A.4 in this revision. Experimental results show that using hysteresis stabilizes learning and improves training performance for most of the models. In addition, training initially failed in MobileNetV2, but using hysteresis enables reliable training, which suggests that hysteresis quantization not only helps the network to reach the optimal point but also prevents divergence in an unwanted direction during the training process.
>
> However, it is interesting to see that the hysteresis quantization is less effective on the LSTM model for the INT4 format. We suspect that this is due to the weight distribution characteristics of the LSTM model. As shown in Fig. 8, most of the weights have a relatively large magnitude in the LSTM model when normalized, contrary to ResNet-18 in which the weights are more evenly distributed. In logarithmic formats, the relative amount of quantization error is similar for all values. In contrast, the relative amount of quantization error is smaller for large values in uniform quantization. Therefore, the weight parameters of LSTM are more severely affected by fluctuation in logarithmic formats, making our hysteresis quantization scheme more effective in those formats compared to uniform quantization. This point is now discussed as our scheme’s limitation in Appendix A.4 in the revised manuscript.
>
> Furthermore, we applied hysteresis quantization to a larger model (ResNet-101), and the result has been added to Table 5. The experimental results confirm that the hysteresis quantization is effective in a large model as well.

---

> ### Author Response · Authors · 2021-11-22
> **ResNet-101 training losses added**
>
> As we promised in our initial response, the training loss of ResNet-101 for each data format has been added to Fig. 3(b) and Fig. 7(b).
>
> FP134, FP143, and FP152 data formats showed training losses of 0.7769, 0.7775, and 0.8187, respectively, which is consistent with the trend predicted by our metrics.

---

> > ### Comment · Reviewer_i9MY · 2021-11-22
> > **Thanks for additional experiments!**
> >
> >
> > I went through responses below and appreciate a lot for additional experiments and comments.
> > Unlike previous works, the major claim in this manuscript is that there exist superior formats (for inference) when we target only a limited number of models. I still believe comparisons seem not to be fair. For example, 8-bit training algorithm (+inference) considers various additional computational techniques to overcome shortcomings (such as losing some intermediate precision) during training while the reasons why such issues would appear are discussed. On the other hand, in this paper, only area overhead and power are considered which might be of interest to the hardware community. Similarly, reference BM8 [4]  also discusses training phase (while this manuscript is handling inference only). Thus, this reviewer still believes that those comparisons are not fair.
> >
> > Since the authors presented detailed responses and additional experiments, I raise the score. But this reviewer still has numerous serious concerns about the motivation of this work.

---

> > > ### Author Response · Authors · 2021-11-23
> > > **We greatly appreciate the reviewer’s effort in reviewing our paper and providing constructive feedback**
> > >
> > > $\textbf{3.	Fair comparisons with other data formats}$
> > >
> > > Since our data format was developed for training, all the experimental results reported in our submission are obtained from training, not inference. In all experiments including ResNet-101 training, the models were trained from scratch with internal variables (weight, activation, error, and weight gradient) deeply quantized into FP134. In other words, those values are represented in FP134 during the entire training process, which is identical to prior works on low-precision training [1-4]. Perhaps the fact that the detailed techniques enabling 8-bit floating-point training were briefly covered in our submission has caused confusion. For better understanding, we compare the training schemes of HFP8, BM8, and FP134 in detail below.
> > >
> > > In HFP8 [3], the authors did not employ the concept of shared exponent bias. Instead, variables are expressed with a fixed exponent bias as in IEEE-754 single-precision and half-precision floating-point formats. By analyzing the statistics of variables in selected models, the authors found that a much larger dynamic range is required for error than activation. Hence, they propose to use different formats for activation and error. However, the data format still had to support a large dynamic range since the activation and error of all layers use a fixed exponent bias.
> > >
> > > On the other hand, BM8 [4] introduces the concept of shared exponent bias into 8-bit floating-point formats. Through this, the data format could provide an optimal dynamic range for values in a block sharing the same exponent bias, enabling reliable training with fewer exponent bits compared to HFP8. The authors propose three techniques for minimizing data loss in the training phase: i) gradual underflow, ii) block size, and iii) hybrid representation. More specifically, they i) implement gradual underflow using subnormal values of floating-point formats, ii) find an optimal block size for exponent bias sharing, and 3) represent activation and error in different formats as in HFP8.
> > >
> > > In this work, we employ the same training scheme as BM8, but improve it further for better processing efficiency. First, we use one exponent bias per layer rather than per block (which is much smaller than a layer) to reduce the storage and tracking overhead of exponent bias. In addition, we use the same data format for activation and error since it will significantly simplify the datapath design. While these two modifications introduce more errors in the training process, we were able to find an optimal FP134 format using the proposed metrics, which minimized the error and resulted in a better training performance than prior works.
> > >
> > > In summary, HFP8, BM8, and our FP134 all quantized activation, weight, error, and weight gradient into 8 bits during training. Our FP134 training with a layer-wise shared exponent bias and unified data format for activation and error achieved higher training performance while also reducing hardware implementation overhead.
> > >
> > > We thank the reviewer again for continuing the discussion and providing valuable feedback so that we can improve our submission. If there are any other aspects of our submission that needs improvements, please feel free to let us know.
> > >
> > > [1] Yang et al., “Swalp: Stochastic weight averaging in low precision training,” ICML 2019
> > >
> > > [2] Cambier et al., “Shifted and squeezed 8-bit floating point format for low-precision training of deep neural networks,” ICLR 2020
> > >
> > > [3] Sun et al., “Hybrid 8-bit floating point (hfp8) training and inference for deep neural networks, “ NeurIPS 2019
> > >
> > > [4] Fox et al,. “A block minifloat representation for training deep neural networks,” ICLR 2020

---

> > > > ### Comment · Reviewer_i9MY · 2021-11-23
> > > > **Thanks for prompt responses**
> > > >
> > > >
> > > > First of all, thanks a lot for the information that training (not inference) is the main target application in this work. But I believe that there could be many reasons why a reader may think that this manuscript does not discuss training aspects with enough details.
> > > > - Figure 1 (or Figure 14) is about inference only. If a paper discusses low-precision training, much more detailed configurations would be required compared to Figure 1 or 14. For example, in ref[2] (S2FP8), figure 4 presents the entire conversion ideas during training to show where format conversions would be made.
> > > > - Low-precision training would need a lot of intermediate results to be stored in different formats. For example, master copy of weights may need additional bits while wide accumulators may be needed to prevent overflow or unnecessary clipping. On the other hand, this paper does not describe any such details. Table 2 and 4 include some bit configurations, but such tables do not involve (potentially necessary) additional hardware resources regarding intermediate results (such as partial sums during matrix multiplications or dot products).
> > > > - Moreover, even though accuracy numbers in the manuscript are good enough, there are no supporting data how we can achieve such good accuracy numbers in the manuscript. For example, all references [1-4] do discuss convergence issues during training that are associated with detailed format conversion issues, rounding methods, formats for master copies, and so on. While Table 2 and 4 may describe the formats for storage of variables, data flow or detailed computation methods (such as accumulation, rounding-off error, and so on) are not presented.
> > > > - Overall, this reviewer feels that while ref[1-4] discuss why and how low-precision training can be achieved (with a lot of detailed techniques to overcome fundamental issues on low-precision training), this paper focuses on the power and area that might be interest of hardware community (even though detailed training operations are not discussed in this manuscript).

---

> > > > > ### Author Response · Authors · 2021-11-24
> > > > > **Thanks again for the careful review**
> > > > >
> > > > > $\textbf{Q4.}$ Overall, this reviewer feels that while ref [1-4] discuss why and how low-precision training can be achieved (with a lot of detailed techniques to overcome fundamental issues on low-precision training), this paper focuses on the power and area that might be interest of hardware community (even though detailed training operations are not discussed in this manuscript).
> > > > >
> > > > > $\textbf{A4.}$ We thank the reviewer for providing insightful feedback. This really helped us understand which point should be clarified further in our submission. Our work and prior studies aim at addressing different issues in low-precision training. In [1-4], the authors look into the cause of training convergence issues or performance drop in the datapath. Then, they try to optimize the datapath by improving rounding methods [3-4] and accumulation schemes [1-4] or introducing loss scaling [3], fixed exponent bias [3], shared exponent bias [2], and hybrid formats [3-4]. However, the data formats for internal variables are still determined somewhat empirically based on the observed statistics.
> > > > >
> > > > > Contrarily, we strive to address the fundamental problem of choosing optimal data formats for those variables. Assuming that the datapath and training schemes (e.g., rounding methods and accumulation precision) are already fixed, our methods allow us to quickly determine optimal data formats that maximize the training performance for the given datapath. While we experimentally verified the effectiveness of our method on the datapath and training schemes presented in [4], it could be applied to other types of datapaths to find data formats optimized for those designs. Furthermore, our method could be extended to datapath optimization. While we only observed the effect of activation and error quantization on the training performance in this work, we could also determine optimal configurations of the datapath such as rounding scheme, accumulation precision, the scope of exponent bias sharing by observing the errors in the weight gradient incurred by those factors. We will include this discussion in the camera-ready version if accepted.
> > > > >
> > > > > Please note that the only reason we compare the area and power consumption of different formats is to show that the proposed FP134 format does not need more hardware resources than prior handpicked data formats. We agree with the reviewer that this part could confuse readers on the main contributions. We will move area/power comparisons to the appendix in the camera-ready version to focus more on the proposed methods and use the additional space to include more details on the training schemes we employed in experiments.
> > > > >
> > > > > [1] Yang et al., “Swalp: Stochastic weight averaging in low precision training,” ICML 2019.
> > > > >
> > > > > [2] Cambier et al., “Shifted and squeezed 8-bit floating point format for low-precision training of deep neural networks,” ICLR 2020.
> > > > >
> > > > > [3] Sun et al., “Hybrid 8-bit floating point (hfp8) training and inference for deep neural networks, “ NeurIPS 2019.
> > > > >
> > > > > [4] Fox et al,. “A block minifloat representation for training deep neural networks,” ICLR 2020.

---

> > > > > ### Author Response · Authors · 2021-11-24
> > > > > **Thanks again for the careful review**
> > > > >
> > > > > $\textbf{Q3.}$ Moreover, even though accuracy numbers in the manuscript are good enough, there are no supporting data how we can achieve such good accuracy numbers in the manuscript. For example, all references [1-4] do discuss convergence issues during training that are associated with detailed format conversion issues, rounding methods, formats for master copies, and so on. While Table 2 and 4 may describe the formats for storage of variables, data flow or detailed computation methods (such as accumulation, rounding-off error, and so on) are not presented.
> > > > >
> > > > > $\textbf{A3.}$ We thank the reviewer for this valuable comment. In this work, we achieved good training accuracy by efficiently finding optimal data formats for internal variables and applying hysteresis quantization. This is independent of the datapath design (e.g., rounding methods, locations of format conversions, and accumulation precision); as we discussed above, we borrowed the datapath and training schemes from BM8 [1] since our goal in this work is not to optimize the datapath itself to relieve performance drop. Instead, we propose a more systematic approach that could be applied to various low-precision training schemes with different configurations. First, we discuss that low-precision training actually consists of two different problems: data flow quantization and network quantization. In typical low-precision training, weight, activation, error, and weight gradient are all deeply quantized. However, we show that quantizing each variable affects the training performance in different ways. For data flow quantization (e.g., quantizing error and activation during training), we experimentally show that the error incurred in the weight gradient is highly correlated with the training performance and propose an efficient method to find an optimal format for those variables. On the other hand, for network quantization (e.g., quantizing weight during training), we show that weight fluctuation is closely related to the performance drop. Based on this observation, we propose a hysteresis quantization scheme to mitigate the fluctuation issue.
> > > > >
> > > > > In summary, our main contribution is not developing an improved training scheme or datapath. Instead, we find fundamental problems in low-precision training that have not been discussed in prior studies and propose a systematic approach consisting of two different methods (optimal data format search and hysteresis quantization) to address those issues. Hence, we believe that these methods could be applied to various training schemes and datapath designs for low-precision training. We will clarify this point in the camera-ready version if accepted.
> > > > >
> > > > > [1] Fox et al., “A block minifloat representation for training deep neural networks,” ICLR 2020.

---

> > > > > ### Author Response · Authors · 2021-11-24
> > > > > **Thanks again for the careful review**
> > > > >
> > > > > $\textbf{Q2.}$ Low-precision training would need a lot of intermediate results to be stored in different formats. For example, master copy of weights may need additional bits while wide accumulators may be needed to prevent overflow or unnecessary clipping. On the other hand, this paper does not describe any such details. Table 2 and 4 include some bit configurations, but such tables do not involve (potentially necessary) additional hardware resources regarding intermediate results (such as partial sums during matrix multiplications or dot products).
> > > > >
> > > > > $\textbf{A2.}$ We thank the reviewer for bringing up an important point. As we discussed above, we exactly followed the training scheme of BM8 [1]. We separately store the high-precision master copy of weights and use high-precision accumulators, as did in [1]. More specifically, as shown in Table 2, BM8 uses a 31-bit fixed-point format for accumulation. Similarly, we use a high-precision accumulator with a 30-bit floating-point format (1-bit sign, 6-bit exponent, and 23-bit mantissa). If we use the same Kulisch accumulator [4] as BM8, our FP134 format would need a 27-bit fixed-point format for lossless accumulation, suggesting that our format incurs lower overhead in the accumulator. For the master copy of weights, BM8 uses a 32-bit floating-point format, but we use a 16-bit floating-point format (1-bit sign, 6-bit exponent, and 9-bit mantissa) as in HFP8 [2]. Also, to mitigate underflow due to a small dynamic range, stochastic rounding [3] is used when quantizing error and weight gradient into 8 bits, as did in BM8.
> > > > >
> > > > > Please note that our approach is different from prior studies on low-precision training. Those studies closely observe data statistics in the target models to determine appropriate formats for internal variables (e.g., weight, activation, error, and weight gradient). Then, they closely investigate the cause of convergence issues during training and try to optimize the datapath to minimize performance drop. Contrarily, our approach is focused on finding an optimal format efficiently through a systematic approach. For a given datapath, our method could find optimal formats for internal variables quickly without actual training. Hence, we believe that our approach complements other low-precision training techniques; although we applied our method to find optimal formats for the datapath in BM8 [1] for experiments in this work, our method could be applied to any other types of datapath to find optimal formats for each design. We will clarify this point in the camera-ready version.
> > > > >
> > > > > [1] Fox et al., “A block minifloat representation for training deep neural networks,” ICLR 2020.
> > > > >
> > > > > [2] Sun et al., “Hybrid 8-bit floating point (hfp8) training and inference for deep neural networks, “ NeurIPS 2019.
> > > > >
> > > > > [3] Wang et al., “Training deep neural networks with 8-bit floating point numbers,” NeuIPS 2018.
> > > > >
> > > > > [4] Kulisch et al., “Computer arithmetic in theory and practice,” Acad. Press 1981.

---

> > > > > ### Author Response · Authors · 2021-11-24
> > > > > **Thanks again for the careful review**
> > > > >
> > > > > We thank the reviewer again for carefully reviewing our response and providing constructive feedback. We have addressed the comments and questions below.
> > > > >
> > > > > $\textbf{Q1.}$ Figure 1 (or Figure 14) is about inference only. If a paper discusses low-precision training, much more detailed configurations would be required compared to Figure 1 or 14. For example, in ref[2] (S2FP8), figure 4 presents the entire conversion ideas during training to show where format conversions would be made.
> > > > >
> > > > > $\textbf{A1.}$ We thank the reviewer for pointing this out. This really helped us track down which part of our submission was misleading and confusing. In Fig. 1(a) and Fig. 14, we wanted to explicitly show where we quantize internal variables in the network for training, not for inference. However, we realized that this could be misinterpreted since only the forward path is depicted in the figures, giving the impression that they only describe the inference process.
> > > > >
> > > > > As pointed out by the reviewer, Fig. 4 in [1] details the datapath and shows where format conversions occur. It also shows that three GEMMs for forward, backward, and weight gradient computation take 8-bit inputs, and the results are quantized back into the same 8-bit format. However, it is unclear how they implemented batchnorm and activation functions. In this work, we exactly followed the training scheme proposed in [2]. As shown in Fig. 3 in [2], all inputs of forward, backward, and weight gradient computation GEMMs are represented in 8 bits, and the results are quantized back into 8 bits after passing through batchnorm and non-linear activation function. We just added a small modification to this scheme: we additionally perform 8-bit quantization before the forward and backward operations of the batchnorm layer to reduce the implementation overhead of the batchnorm layer. Our training scheme is detailed below, along with comparisons with BM8 [2].
> > > > >
> > > > > a)	BM8 [2]
> > > > >
> > > > > &nbsp;&nbsp;&nbsp;&nbsp;FP32 to FP8 $\rightarrow$ forward GEMM $\rightarrow$ batchnorm or activation function $\rightarrow$ FP32 to FP8
> > > > >
> > > > > &nbsp;&nbsp;&nbsp;&nbsp;FP32 to FP8 $\rightarrow$ backward GEMM $\rightarrow$ backward function of batchnorm or activation function $\rightarrow$ FP32 to FP8
> > > > >
> > > > > &nbsp;&nbsp;&nbsp;&nbsp;FP32 to FP8 $\rightarrow$ weight gradient GEMM $\rightarrow$ FP32 to FP8 $\rightarrow$ weight update to master copy
> > > > >
> > > > > b)	Ours
> > > > >
> > > > > &nbsp;&nbsp;&nbsp;&nbsp;FP32 to FP8 $\rightarrow$ forward GEMM $\rightarrow$ activation function $\rightarrow$ FP32 to FP8 $\rightarrow$ batchnorm (if it exists) $\rightarrow$ FP32 to FP8
> > > > >
> > > > > &nbsp;&nbsp;&nbsp;&nbsp;FP32 to FP8 $\rightarrow$ backward GEMM $\rightarrow$ backward function of activation function $\rightarrow$ FP32 to FP8 $\rightarrow$ backward function of batchnorm (if it exists) $\rightarrow$ FP32 to FP8
> > > > >
> > > > > &nbsp;&nbsp;&nbsp;&nbsp;FP32 to FP8 $\rightarrow$ weight gradient GEMM $\rightarrow$ FP32 to FP8 $\rightarrow$ weight update to master copy
> > > > >
> > > > > While this point is briefly explained in Section 2.2, we agree with the reviewer that more details should be provided in our manuscript so that a reader can understand details without referring to other papers. We will update those figures and add more information in the camera-ready version if accepted.
> > > > >
> > > > > [1] Cambier et al., “Shifted and squeezed 8-bit floating point format for low-precision training of deep neural networks,” ICLR 2020.
> > > > >
> > > > > [2] Fox et al,. “A block minifloat representation for training deep neural networks,” ICLR 2020.

---

> > > ### Author Response · Authors · 2021-11-23
> > > **We greatly appreciate the reviewer’s effort in reviewing our paper and providing constructive feedback**
> > >
> > > We greatly appreciate the reviewer’s effort in reviewing our paper and providing constructive feedback. Please see below for our responses to the reviewer’s questions and comments.
> > >
> > > First of all, it seems that some part of our manuscript or previous responses was misleading and has caused misunderstandings. We apologize for the confusion. If the reviewer could point us to those parts, we will try our best to improve our submission through revision or in the camera-ready version if accepted since we are very close to the deadline for uploading a revised manuscript.
> > >
> > > $\textbf{1.	Our data format is for training, not for inference}$
> > >
> > > In this work, we are proposing a method to find optimal data formats for training (not inference). We suspect that our claim that we could find an optimal format without actual training has caused serious confusion. This does not mean that our data format is aimed at inference. We are still trying to find an optimal data format for training, but we could compare the expected training performance of various data formats without actually training all different models using those formats. More specifically, a typical design procedure for determining an optimal data format for low-precision training is as follows:
> > >
> > > &nbsp;&nbsp;&nbsp;a)	 Select representative models for various tasks such as image classification, translation, and object detection
> > >
> > > &nbsp;&nbsp;&nbsp;b)	 Observe statistics of internal variables such as activation, errors, and weight gradients in those models
> > >
> > > &nbsp;&nbsp;&nbsp;c)	Select data format candidates considering the required dynamic range and precision of each variable
> > >
> > > &nbsp;&nbsp;&nbsp;d)	Train the selected models from scratch using each data format and measure training performance
> > >
> > > &nbsp;&nbsp;&nbsp;e)	Choose the data format with the best training performance
> > >
> > > In this procedure, step d) takes most of the time and computations since it requires end-to-end training of all the models in various data formats. However, using our metrics, we could simplify the steps d) and e) as follows:
> > >
> > > &nbsp;&nbsp;&nbsp;d)	Apply each data format to quantize activation and error (gradient of activation) in pre-trained full-precision models and observe errors in the calculated weight gradient for a small training set
> > >
> > > &nbsp;&nbsp;&nbsp;e)	Choose the data format with the smallest weight gradient error
> > >
> > > Our method significantly reduces the amount of simulation time and computational resources required in step d) since now we only need to observe the amount of error introduced in the weight gradient after applying different data formats. For instance, if we need to compare the training performance of 100 data formats, we need to train a model from scratch 100 times while applying a different data format in each run in a conventional approach. Contrarily, in our method, we quantize the activation and error in a pre-trained model into a different format and observe the error in weight gradients for a small set of training data (100 mini-batches in our experiments). Hence, the amount of time to estimate the training performance of each format is significantly reduced, which allows for finding an optimal data format from a large set of possible data formats.
> > >
> > > $\textbf{2.	We do not target a limited number of models but aim to find a data format suitable for training a wide range of models }$
> > >
> > > As discussed above, the proposed method allows us to predict the training performance of various data formats. Using this method, we aim to find an optimal format for training a wide range of models in this work. Following the approaches in prior works [3-4], we first choose the representative models of various tasks and apply our method to find the best data format. We found that FP134 exhibits the smallest weight gradient errors (i.e., best training performance) and showed that this FP134 format achieves a higher training performance than other data formats handpicked for low-precision training. Since our format has been verified on a similar or larger number of models compared to prior works [1-5], we believe that the FP134 format is also suitable for training a wide range of models. In the first revision, we wanted to conservatively point out that our format was tested on a specific set of models (just like any other prior studies on low-precision training), but it was somehow misinterpreted and caused confusion.
> > >
> > > [1] Yang et al., “Swalp: Stochastic weight averaging in low precision training,” ICML 2019
> > >
> > > [2] Cambier et al., “Shifted and squeezed 8-bit floating point format for low-precision training of deep neural networks,” ICLR 2020
> > >
> > > [3] Sun et al., “Hybrid 8-bit floating point (hfp8) training and inference for deep neural networks, “ NeurIPS 2019
> > >
> > > [4] Fox et al,. “A block minifloat representation for training deep neural networks,” ICLR 2020
> > >
> > > [5] K ̈oster et al., “Flexpoint:  An adaptive numerical format for efficient training of deep neural networks,” NeurIPS 2017.

---

### Official Review · Reviewer_AC1G · 2021-11-09

**Correctness:** 3
**Technical Novelty And Significance:** 3
**Empirical Novelty And Significance:** 3
**Recommendation:** 8
**Confidence:** 5

**Main Review:**

While there are few nitpicks I have (these should be fixed prior to publishing), the ideas in the paper are well-founded and useful.  I believe there is a meaningful contribution to the field here.

In the intro, there really needs to be a citation for Koster et al (2017) on Flexpoint. That was one of the first papers discussing these issues and one of first implementations of different numerics in modern hardware.

In the second paragraph of the introduction, the line "When a value is represented using a fixed number of bits, there is a trade-off between dynamic range and precision" is not quite correct.  This is only true for floating-point formats, so simply stating that will clear this up.

In the "Performance degradation in from-scratch training" section: The line "the neural network is trained from scratch while all the values in the network - not only parameters, but also other variables in the network (e.g., activation, error, and weight gradient) " is not necessarily true and different components are done differently even in this paper. I'm not sure why there is so much text devoted to differentiating quantized vs from-scratch training, but I think this could removed.  The paper is about from-scratch training so just state that.

Under Numeric Formats, I'm not sure what a "symmetric" format is and why 2's complement is asymmetric.  Please define or remove.

Under Effect of quantized activation, it says "...suggesting that the misalignment angle is a better metric to predict the ranking of various formats based on the training performance." I think this is true, but more due to the shape of the curve rather than the (small) effect on spearman correlations.  It might be clearer to state this; the angle differences are high for even small losses.

The hysteresis method has some great results and might really be the key to unlocking lower precision.  Great work!

**Summary Of The Paper:**

The authors propose 2 impactful methods to aid in the design of numeric precision for neural network training: a method to quickly determine which formats work for weights and activations using angular deviation of gradients between low precision and FP32, and a hysteresis method for dealing with low precision representations.

**Summary Of The Review:**

The paper presents 2 very important ideas that I believe are of high utility to low precision efforts. The idea of weight hysteresis could actually spawn many new methods that lead to better hardware going forward.

---

> ### Author Response · Authors · 2021-11-17
> **Response to "Official Review 1"**
>
> We thank the reviewer for carefully reviewing our submission. We have addressed all questions and comments in the below section and made appropriate changes to the manuscript in this revision.
>
> $\textbf{Q1}$. In the intro, there really needs to be a citation for Koster et al (2017) on Flexpoint. That was one of the first papers discussing these issues and one of first implementations of different numerics in modern hardware.
>
> $\textbf{A1}$. We thank the reviewer for the suggestion. We have cited the paper in the introduction section of the revised manuscript.
>
> $\textbf{Q2}$. In the second paragraph of the introduction, the line "When a value is represented using a fixed number of bits, there is a trade-off between dynamic range and precision" is not quite correct. This is only true for floating-point formats, so simply stating that will clear this up.
>
> $\textbf{A2}$. We thank the reviewer for pointing this out. We have corrected the sentence to show that this only applies to floating-point formats.
>
> $\textbf{Q3}$. In the "Performance degradation in from-scratch training" section: The line "the neural network is trained from scratch while all the values in the network - not only parameters, but also other variables in the network (e.g., activation, error, and weight gradient) " is not necessarily true and different components are done differently even in this paper. I'm not sure why there is so much text devoted to differentiating quantized vs from-scratch training, but I think this could be removed. The paper is about from-scratch training so just state that.
>
> $\textbf{A3}$. We thank the reviewer for this valuable feedback. In the original manuscript, we wanted to clarify the difference between the quantized model and from-scratch training. However, as pointed out by the reviewer, this part was lengthy and included inaccurate explanations. We have simplified this discussion in the revised manuscript.
>
> $\textbf{Q4}$. Under Numeric Formats, I'm not sure what a "symmetric" format is and why 2's complement is asymmetric. Please define or remove.
>
> $\textbf{A4}$. We apologize for the confusion. Symmetric formats have identical representable ranges for positive and negative numbers. For instance, an N-bit two’s complement number format has the representable range of $-2^{k-1}$ to $2^{k-1}-1$, suggesting that the format can represent one more negative number than the positive ones. Hence, the two’s complement format is asymmetric. This point is now clarified in Section 2.1 in the revised manuscript.
>
> $\textbf{Q5}$. Under Effect of quantized activation, it says "...suggesting that the misalignment angle is a better metric to predict the ranking of various formats based on the training performance." I think this is true, but more due to the shape of the curve rather than the (small) effect on spearman correlations. It might be clearer to state this; the angle differences are high for even small losses.
>
> $\textbf{A5}$. We thank the reviewer for this insightful feedback. This is a very valid point. While these two metrics exhibit similar Spearman’s correlation, the shape of the loss-error curve is more distinct for small losses when using the misalignment angle, making it a better metric for format selection. This information has been included in Section 2.3 in this revision.

---

### Decision · Program_Chairs · 2022-01-20

**Decision:**

Accept (Poster)

**Comment:**

This paper introduces a method to determine which precision to use for the weights, as well as a quantisation method using hysteresis to improve performance with low-precision weights, including 4-bits.
Reviewers tend to agree that the two points presented are useful and can have a large impact on the field.
Generally, reviewers pointed out that motivations, notations and experimental studies could be improved. This has been partly addressed by the authors.
I recommend to accept this paper for ICLR 2022.